# Broadband random optoelectronic oscillator

Zengting Ge[1,2], Tengfei Hao[1,2], José Capmany [3], Wei Li[1,2], Ninghua Zhu[1,2] & Ming Li [1,2 ✉]

Random scattering of light in transmission media has attracted a great deal of attention in the field of photonics over the past few decades. An optoelectronic oscillator (OEO) is a microwave photonic system offering unbeatable features for the generation of microwave oscillations with ultra-low phase noise. Here, we combine the unique features of random scattering and OEO technologies by proposing an OEO structure based on random distributed feedback. Thanks to the random distribution of Rayleigh scattering caused by inhomogeneities within the glass structure of the fiber, we demonstrate the generation of ultra-wideband (up to 40 GHz from DC) random microwave signals in an open cavity OEO. The generated signals enjoy random characteristics, and their frequencies are not limited by a fixed cavity length figure. The proposed device has potential in many fields such as random bit generation, radar systems, electronic interference and countermeasures, and telecommunications.

[1] State Key Laboratory on Integrated Optoelectronics, Institute of Semiconductors, Chinese Academy of Sciences, 100083 Beijing, China. [2] School of Electronic, Electrical and Communication Engineering, University of Chinese Academy of Sciences, 100049 Beijing, China. [3] Photonics Research Labs, ITEAM Research Institute, Universitat Politecnica de Valencia, Camino de Vear s/n, 46022 Valencia, Spain. ✉email: ml@semi.ac.cn

The propagation of light in random media has attracted a great deal of attention in the past few decades, rendering a plethora of developments in biomedical imaging, sensing, and optical communication fields[1–4]. Several studies on the effects of light propagation in random media have concluded that it is possible to take advantage of the intrinsic random disorder in the material. As one of the highlight phenomena, Rayleigh scattering originating from inhomogeneities within the glass structure of the fiber has a huge potential in practical applications, such as random lasers[5,6] and optical time-domain reflectometry[7]. On the other hand, photonic-assisted microwave generation systems have been shown to be outstanding in generating higher frequency and lower phase noise microwave signals. An optoelectronic oscillator (OEO) is a particularly efficient scheme, which includes a positive feedback loop to create microwave oscillation and generate a single-frequency microwave signal with ultra-low-phase noise[8]. Various architectures with increased refinement and complexity have been proposed and experimentally demonstrated along the last years providing remarkable progress in terms of performance. Among these, we can highlight coupled, multi-loop, injection-locked, Fourier domain mode-locked (FDML), monolithically integrated and parity-time-symmetric OEOs[9–15]. In essence, ordinary OEOs are based on closed fiber cavities, which feature a very limited frequency oscillation range centered around a discrete set of cavity modes with oscillating frequencies determined by the fixed cavity length[8]. Many attempts on the different OEO structure have been reported over the past years in order to increase the operating frequency range and reduce phase noise. All these contributions rely on fixed close cavity structures (single or dual loop).

Here, we leverage on the combination of Rayleigh backscattering mechanism with an OEO device to propose an optoelectronic system with unique features beyond the state of the art in OEO technology. In the proposed device the feedback signal is not obtained by means of a cavity, as the loop is left open. Rather, it generates from the Rayleigh backscattering provided by an optical fiber. Leveraging on Rayleigh backscattering, one can obtain a virtually (i.e., not physically) closed random distributed feedback signal while keeping an open-loop physical cavity. In this paper, we demonstrate an open cavity broadband random OEO based on this random distributed feedback, producing broadband random signals. The proposed broadband random OEO brings the unique ability to generate ultra-wideband random signals with the added value of relying on a simple structure not requiring precise cavity length adjustments.

## Results

**Principle.** The block diagram of the broadband random OEO is shown in Fig. 1. A laser diode (LD) is used to generate a continuous lightwave signal, which is sent to a Mach–Zehnder modulator (MZM). The output is coupled into a dispersion compensation fiber (DCF) through a wavelength division multiplexer (WDM) together with a Raman pump lightwave signal. Rayleigh scattering and distributed Raman amplification occur while the signal and pump light propagate along the optical fiber. Due to limitations in the fiber manufacturing process, its internal refractive index of fiber is not uniform, leading to inherent Rayleigh scattering that is produced in all directions, among which, the weak backscattering is recaptured by the fiber propagating in the opposite direction to the incident light in the fiber[16,17]. Distributed Raman amplification is used as the gain mechanism to amplify both the incident and backscattered light. The DCF is used to provide a larger Rayleigh backscattering coefficient due to its higher fiber loss, and is angled cleaved at the open-end facet to eliminate Fresnel reflection to make sure that

the feedback comes from the Rayleigh scattering only. The backscattered part of signal light propagates in the opposite direction to incident light, experiencing further amplification in the erbium-doped fiber amplifier (EDFA) placed after Port 3 of the optical circulator. Since Raman amplification has a large gain bandwidth, we use a narrow-band optical filter to select the signal light and suppress the energy of other wavelengths. The optical signal is then converted to an electrical signal in a photodetector (PD). The microwave signal is then amplified through an electric amplifier (EA) and filtered by a band-pass radiofrequency filter. A microwave power splitter is used to split a fraction of the microwave signals to a spectrum analyzer and a real-time oscilloscope for analysis. The remainder of the signal is injected to the MZM to virtually close the loop.

The conditions for coherent addition and a loop gain exceeding losses are essential for OEO to be able to self-oscillate. For the first condition, the oscillation frequencies in a conventional single-loop OEO are a discrete set given by:

$$f_{osc} = \frac{mc}{nL}, \qquad (1)$$

where $m$ is an integer value, $n$ is the refractive index, $c$ is the velocity of light in vacuum, and $L$ is the length of the cavity. Adjacent mode separation is given by $\Delta f_{osc} = \frac{c}{nL}$.

For the second, the oscillation threshold of the OEO is given by:

$$V_{ph} = V_\pi / [\pi\eta|\cos(\pi V_B/V_\pi)|], \qquad (2)$$

where $V_\pi$ and $V_B$ are the half-wave voltage and the bias voltage of the modulator, respectively, $\eta$ is the extinction coefficient of the modulator, and $V_{ph}$ is the photovoltage. With precise cavity length and adequate gain that satisfies the threshold, the desired frequencies can oscillate from the background noise floor in the closed loop.

In the proposed broadband random OEO, Rayleigh backscattering from an open fiber is used to provide feedback and close the loop. Due to the random density fluctuations caused by limitations in the fiber fabrication, Rayleigh scattering occurs at statistically independent spatial positions along the fiber from the input ($z_0$) to the open port ($z_L$), and only a fraction of the scattered light is recaptured by the fiber at scattering angles close to $\pi$ compared to that propagating in the opposite direction. It is worth mentioning that the instantaneous Rayleigh scattering amplitude at each scatter section fluctuates randomly in time, with a Gaussian distribution[18–20]. A backscattering coefficient $\Delta\rho$ ($z_i$) is used to measure the fraction of the scattered field from a section, and the coefficient $\Delta\rho(z_i)$ is a time-independent zero-mean circular complex Gaussian random variable. The backscattered field at the input of the DCF from a scatter section $z_i$ is given by:

$$\Delta\varepsilon_b(t, z_i) = M(z_i)\varepsilon_s(t - 2z_i/v)e^{-\alpha z_i}e^{-j2\beta z_i}\Delta\rho(z_i), \qquad (3)$$

where $\alpha$ is the attenuation coefficient, $\beta$ is the propagation constant, $v$ is the group velocity, and the unitary Jones matrix $M(z_i)$ describes the polarization state.

A large amount of Rayleigh backscattering can be thought of a sum of gratings inscribed at different positions in the fiber, these, in turn, can be interpreted as implementing different feedback cavity lengths. Thus we can regard the broadband random OEO as a sum of single loop OEOs with different, and randomly distributed cavity lengths, $L_i = L_0 + 2z_i$, where $z_i$ is the coordinate position of backscattering inhomogeneity in the fiber and $L_0$ is the length of OEO cavity excluding the DCF. For any frequency $f$, there are a set of cavity lengths $L_i$ ($L_i = \frac{m\lambda}{n}$, $\lambda$ is the corresponding wavelength of frequency $f$) that meet the phase matching condition. The total backscattered field superimposed

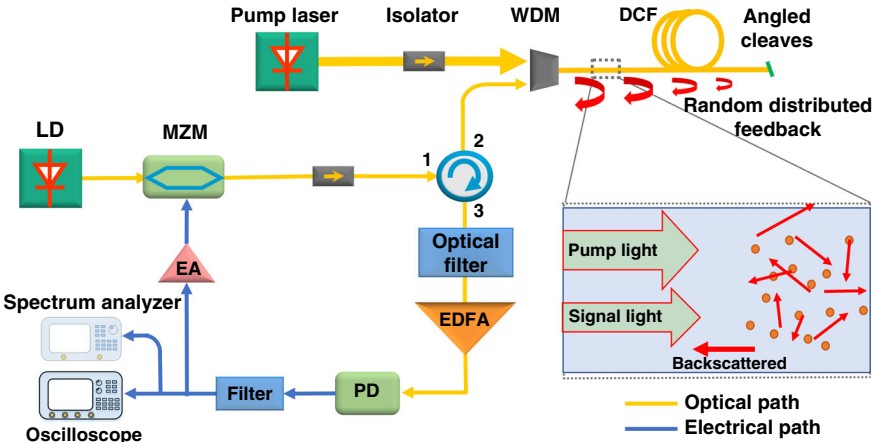

**Fig. 1 Block diagram of the broadband random OEO.** LD, laser diode; MZM, Mach–Zehnder modulator; WDM, wavelength division multiplexer; DCF, dispersion compensation fiber; EDFA, erbium-doped fiber amplifier; PD, photodetector; EA, electrical amplifier. Angled cleaving is used at the fiber end facets.

from all these statistically independent sections is $\varepsilon_b(t) = \sum_{n=1}^{N} \varepsilon_b(t, n\Delta l)$, also showing a random fluctuation property. Hence, the total gain in the ring cavity cannot be kept constant, resulting in the phenomenon that the power of the oscillating signal displays obvious random fluctuations.

Waves of different frequencies always find the corresponding cavity lengths to meet the coherent addition condition in this physical ring cavity. Thus, the oscillation frequencies $f_{osc}$ are continuous without any discrete mode interval ($\Delta f_{osc} \approx 0$). When the gain in the cavity is large enough, all these waves that satisfy the coherent addition condition start to oscillate simultaneously. All these Rayleigh backscattering at random distributed positions result in a continuous change in signal frequencies and a random fluctuation in signal power, rather than a single or discrete set of fixed values, making this device completely different from conventional OEOs reported so far.

## Experiment

The proposed OEO was assembled and experimentally tested (see methods). The obtained signal results when a radiofrequency filter centered at 5 GHz was employed to select the output spectrum are shown in Fig. 2. Figure 2a shows the measured spectrum of the microwave signals oscillating in the OEO cavity. As can be seen, there is a continuum of frequencies in the passband oscillating in the loop without any appreciable longitudinal mode interval. The inset on the right side shows a portion of the spectrum with a span of 600 kHz centered at 5 GHz, which clearly shows that there are no discrete longitudinal modes under different spans. Here a radiofrequency filter centered at 5 GHz with a 3-dB bandwidth of 60 MHz was used in the cavity in order to clearly distinguish the noise floor and the oscillation frequencies by the difference in power. The ramp at the top of the spectrum is due to the non-uniformity in the magnitude response of the radiofrequency filter. The frequency response curve of the filter is measured and put as an inset in Fig. 2a. It can be seen that the response curve at the center of the filter is not flat enough. The unevenness of the frequency response curve will be further amplified by processes such as gain and mode competition in the optoelectronic link, and the unevenness will appear in the spectrum result. If a filter with a flatter frequency response is used, a random signal with a flat spectrum can be generated.

The temporal waveform of the output microwave signal from the splitter is sampled with a real-time oscilloscope at a sampling rate of 100 GS/s as shown in Fig. 2b. The amplitude varies

randomly, showing a probability density function (PDF, blue solid line) of Gaussian distribution, with the mathematical expectation μ is $-7.9 \times 10^{-3}$, and the standard deviation σ is 0.1451, as shown in the right panel in Fig. 2b. We also calculated the ideal Gaussian distribution (orange dotted line), and calculated the correlation coefficient between the probability density function and the ideal Gaussian distribution. The value of the correlation coefficient $R$ is 0.9979. The deviation of the probability density distribution may be caused by the noise introduced by other devices in the link and the unevenness of the oscillating signal in the passband. The experimental results are relatively consistent with the theory[18]. The inset shows the details of the waveform, which approximates a sinusoidal signal with the period varies slightly around 0.2 ns.

A short-time Fourier transform was performed to the sampled data. In this process, a Hamming window was selected to intercept the signal, and an appropriate window length was selected. In this paper, the window length is set to 6 μs. Two adjacent windows have an intersection, and the length of the overlapping part is set to 3 μs. The result is shown in Fig. 2c. It can be appreciated that the temporal waveform of the signal is a superposition of a small continuum of frequencies rather than a periodic single-frequency signal. All the frequencies contained in the passband of the radiofrequency filter oscillate in OEO at the same time. For each couple of adjacent time windows, the power of a particular frequency changes randomly, and within a given time window, the power varies at different frequencies. This power-varying phenomenon is a consequence of the random nature of Rayleigh scattering. The superimposed backscattered light fields from different, statistically independent sections of the fiber fluctuate randomly in time and these fluctuation characteristics are converted from the optical to the microwave spectrum in the PD[21]. Randomness is an inherent characteristic of the down-converted radiofrequency signal, the corresponding power for each frequency component is different since each one is the result of a different time-varying scattering process.

We then carried out the simulation of the broadband random OEO. Based on the aforementioned assumptions, a bandpass radiofrequency filter centered at 5 GHz with a bandwidth of 60 MHz is simulated with reference to the experiment. Considering the huge amount computation power required for an accurate simulation, we only take the first 0.6 km of the DCF fiber in the simulation where the Ramon gain is most significant, corresponding to a total cavity delay between $6 \times 10^{-6}$ s and $6 \times 10^{-8}$ s, although a total of 10 km DCF fiber was used in the experiment.

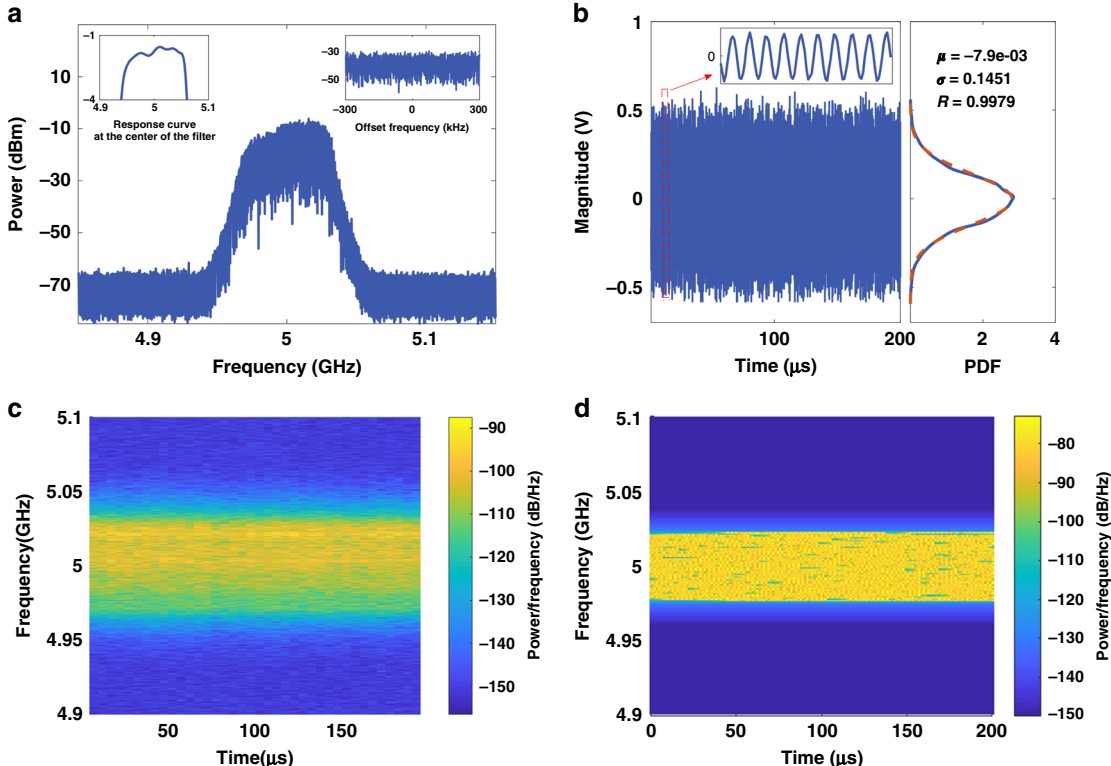

**Fig. 2 Experimental results.** A radiofrequency filter centered at 5 GHz with a 3-dB bandwidth of 60 MHz is used in the cavity. **a** Spectrum results of signals oscillating in the OEO. All frequencies in the passband oscillate in the OEO without discrete mode interval. The inset on the right shows a section of the spectrum with a span of 600 kHz, which clearly shows that there are no longitudinal modes under different spans. The inset on the left displays the response curve near the center frequency of the filter. **b** Temporal waveform and the probability density function of the signal. The inset is a section of the waveform. The waveform approximates a sinusoidal signal with a period that varies slightly around 0.2 ns. The right panel displays the Gaussian probability density function of the random signal as shown by the blue solid line. The orange dotted line shows the Gaussian distribution. The mathematical expectation $\mu$ is $-7.9 \times 10^{-3}$, and the standard deviation $\sigma$ is 0.1451. The correlation coefficient R is 0.9979. **c** Real-time frequency distribution of the temporal waveform. The temporal waveform is a superposition of all oscillation frequencies in the passband rather than a periodic single-frequency signal. All the frequencies contained in the passband of the radiofrequency filter oscillate in OEO at the same time. For each couple of adjacent time windows, the power of a particular frequency changes randomly, and within a given time window, the power varies at different frequencies. **d** Simulation result of the frequency distribution.

The OEO model including the contribution of additive noise in the cavity follows the general equations developed in Refs. [22,23]. In additions, random characteristics of Rayleigh backscattering are incorporated into this model. In this simulation, the statistical properties of backscattered light are first down-converted from optical to the microwave domain in the PD and then up-converted from the microwave to the optical spectrum in the MZM. After dozens of iterations, the initial random field is amplified and filtered through the optoelectronic link connecting the PD and the MZM. Figure 2d shows the simulation results of the broadband random OEO. Note that all the frequencies contained in the passband of the radiofrequency filter oscillate in OEO at the same time. The power fluctuates over time, which is consistent with the experimental results shown in Fig. 2c.

The proposed structure is capable of generating ultra-wideband random microwave signals by simply tuning the central frequency and bandwidth of the radiofrequency filter. This is a remarkable and unique characteristic of this device since ordinary OEOs based on closed fiber cavities have a very limited frequency oscillation range always center around a discrete set of cavity modes. To demonstrate this feature, we changed the RF filter characteristics from the 5 GHz (C band) bandpass response to a low-pass characteristic with a cutoff frequency of 3.3 GHz (S and L bands) and then to a bandpass filter centered at 12 GHz (X Band) with a 3-dB bandwidth of 200 MHz. The results of the

broadband low-pass spectrum and the narrow-band bandpass spectrum are shown in Fig. 3.

As can be seen in both Fig. 3a, b, all the frequencies in the passband can oscillate in the OEO cavity. The power of oscillating frequencies is much higher than the background noise floor. With the proposed OEO structure, broadband frequencies can be generated in the OEO, and thus the oscillating modes can be tuned to different bands enabling different services. The broadband random OEO can therefore be applied in service scenarios requiring wideband, frequency tunable feature, such as 5/6 G communications, radar, and autonomous driving, which greatly reduces the complexity of frequency source devices.

A key characteristic related to the available RF spectrum is that the device is capable of providing ultra-wideband oscillation. This is obtained by measuring the output signal spectrum without any radiofrequency filter in the link as shown in Fig. 4a. As it can be appreciated, for the particular configuration assembled in the laboratory, the power of the oscillation signals is about 20 dB higher than the noise floor from DC to about 40 GHz, indicating that all the frequencies within this range oscillate in the broadband random OEO. This feature is, again, unique from this device and would allow, for instance, the generation of ultra-wideband for 5 G communications within the S and C bands (2–6 GHz), weather monitoring, air traffic control, maritime vessel traffic control and defense tracking radar applications within the X band

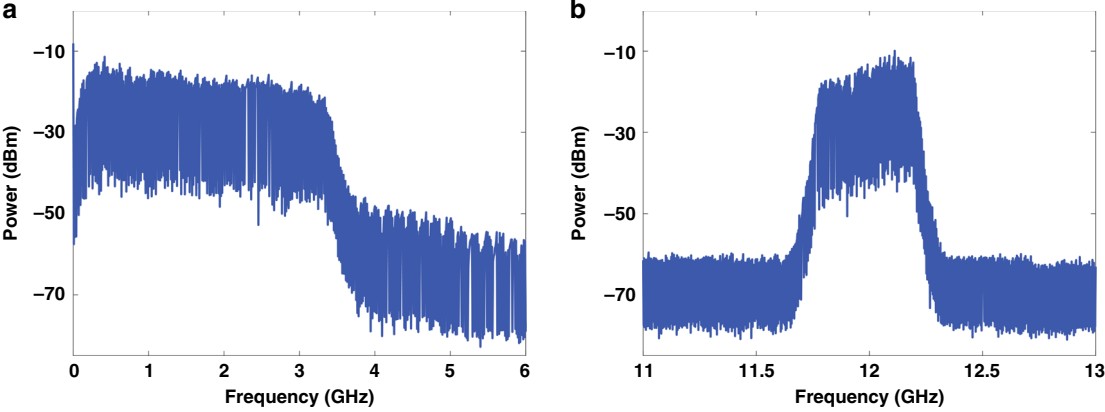

**Fig. 3 Spectrum results with different radiofrequency filters in the link. a** A low-pass radiofrequency filter with the cutoff frequency at 3.3 GHz (S & L bands) is placed in the cavity. **b** A bandpass radiofrequency filter centered at 12 GHz (X band) with a 3-dB bandwidth of 200 MHz is placed in the cavity.

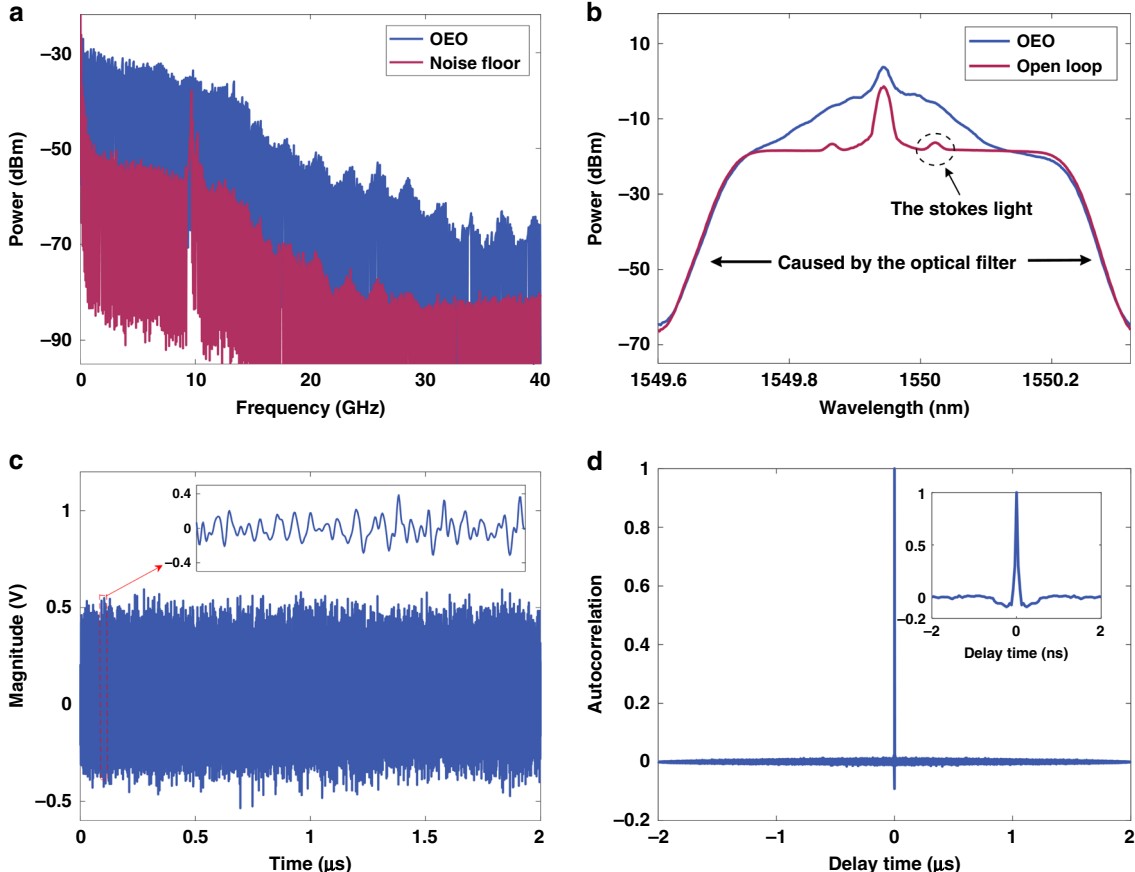

**Fig. 4 Experimental results without any radiofrequency filter in the link. a** The broadband signal spectrum result. Blue: spectrum result of oscillating frequencies in the OEO. Red: the noise floor without feedback to the MZM. All the frequencies within the passband (from DC to about 40 GHz) oscillate in the broadband random OEO. The power of the oscillation signals is about 20 dB higher than the noise floor. **b** Optical spectrum measured after the EDFA. Blue: optical spectrum in a closed loop. Red: optical spectrum in open loop (without feedback to the MZM). The optical spectrum is significantly broadened when the loop is closed, and all frequencies of the broad spectrum are modulated onto the signal light. **c** Temporal waveform from the OEO. The inset shows a time window of the waveform. No periodicity is seen from the temporal waveform, and its amplitude varies randomly. **d** Autocorrelation function. The inset shows the expanded view of the center peak. The autocorrelation function has a maximum value at delay time $\tau = 0$ and decays to zero quickly when delay time $\tau$ is slightly greater than zero, confirming that the generated signal is non-periodic and has approximately white noise characteristics.

(8–12 GHz), satellite data links in the Ku-band (12–18 GHz) and K-band (18–25 GHz), future evolutions of 5 G and 6 G telecommunication systems and next-generation broadband wireless line-of-sight links using carriers from 28 to 35 GHz, etc. Furthermore, the oscillation power could be higher with larger

optoelectronic device's maximum output and input RF power. The higher noise floor power is because more active devices such as amplifiers are used in the link, which introduces additional noise. The high noise floor of the link is mainly due to the use of many active devices, such as amplifiers, which introduce

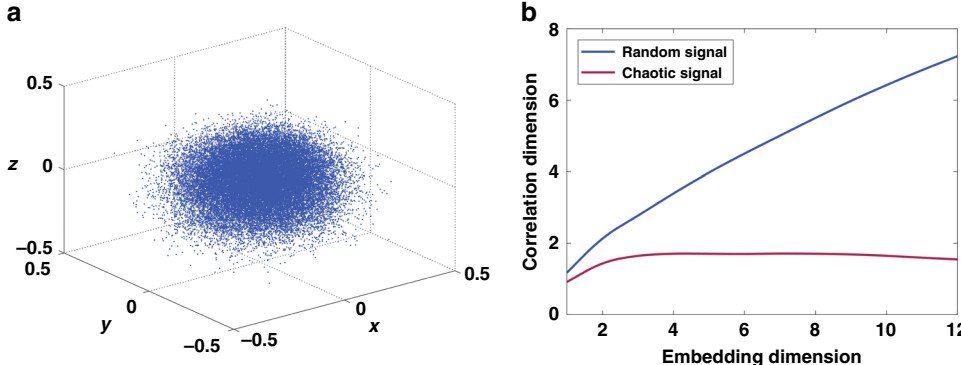

**Fig. 5 The phase space and correlation dimension results. a** The phase space of signal generated in the broadband random OEO, presenting a scatter plot. **b** The relationship between correlation dimensions and embedding dimensions of a chaotic waveform and a random waveform sampled from the broadband random OEO. Correlation dimension value tends to saturate with the increase of the embedding dimension for chaotic time series, while for random time series the correlation dimension value does not saturate.

| Table. 1 The difference between the conventional single-frequency OEO, the chaotic OEO and the broadband random OEO. | | | |
|---|---|---|---|
| **Classification** | **Conventional single-frequency OEO** | **Chaotic OEO** | **Broadband random OEO** |
| Feedback mechanism | Closed loop | Closed loop | Open loop and Rayleigh scattering |
| Signal Classification | Deterministic and periodic signal | Deterministic and aperiodic signal | Stochastic and aperiodic signal |
| Oscillation frequency range | Single frequency (in the tunable range) | Broadband | Broadband (All frequencies within the passband) |
| Signal complexity | Simple | Complex | Complex |

additional noise. We need to use an optical amplifier to amplify the weaker Rayleigh scattered light, which raises the link noise floor. A low-noise amplifier can be used to reduce the noise floor of the link.

A narrow peak at ~ 9.8 GHz is shown in the noise floor in Fig. 4a due to the Stokes shift of the stimulated Brillouin scattering (SBS). This peak disappears in the microwave signal spectrum, because spectral broadening increases the threshold of SBS[24]. The spectral broadening is displayed. The pedestal in the lower part of the spectrum in Fig.4b is caused by the narrow-band optical filter used in the experiment to filter out other wavebands. The two small peaks on the left and right sides of the optical carrier in the red line are the Stokes light and the anti-Stokes light generated by the SBS process. It can be seen from Fig. 4b that the optical spectrum is significantly broadened when the loop is closed, and all frequencies of the broad spectrum are modulated onto the signal light.

The time-domain waveform of signal was also measured with a real-time oscilloscope at a sampling rate of 100 GS/s and results are shown in Fig. 4c, where the inset shows a time window of the waveform. No periodicity can be seen from the temporal waveform, and its amplitude also varies randomly. We calculated the autocorrelation function of the signal, and the result is approximately a delta function, as shown in Fig. 4d. The autocorrelation function has a maximum value at delay time $\tau = 0$ and decays to zero quickly when delay time $\tau$ is slightly greater than zero. (The negative value is because the absolute value of the autocorrelation result is not taken, and in some papers, the absolute value of the autocorrelation function is taken to facilitate the comparison of side lobes.) This autocorrelation confirms that the generated signal is non-periodic and has approximately white noise characteristics. The time delay of the cavity cannot be deduced from this result, and the signals have excellent unpredictability, indicating that the proposed scheme can be used as microwave photonic noise source[25]. It should be pointed out that the

principle to generate a random spectrum for the proposed OEO is not similar to an amplified noise source, such as amplified spontaneous emission (ASE). ASE generates optical noise by optically amplifying the spontaneous radiation in the gain medium. While the broadband random OEO combines Rayleigh scattering to form a large number of random cavities, and waves satisfying the coherent superposition conditions can self-oscillate in these cavities, thereby generating random signals.

However, the non-periodic and irregular characteristics of a signal generated in the broadband random OEO is similar to chaos. Different from the traditional single-frequency OEO, another type of OEO, i. e., chaotic OEO has been used as a delayed feedback system to generate nonlinear dynamic behaviors such as chaos through the nonlinear effect in the cavity[26–28]. In chaotic OEO, an electro-optic modulator (EOM) is used as the core nonlinear device. By changing the feedback strength, the system enters an unsteady state and generates a complex chaotic signal. As a deterministic signal, a chaotic signal has the characteristics of non-periodic, irregular and similar to white noise. To further distinguish the random waveform from a chaotic waveform, it is necessary to reconstruct the phase space and calculate the relationship between the correlation dimension and the embedding dimension in the reconstructed phase space. In phase space, periodic time series become a closed orbit and chaotic time series reconstruct into fractal orbits whose orbital segments exhibit sensitive dependence on initial conditions, and random series presents a scatter plot. The correlation dimension value tends to saturate with the increase of the embedding dimension for chaotic time series, while for random time series the correlation dimension does not saturate.

We reconstruct the phase space with the time series of the waveform sampled from the broadband random OEO by using Wolf's algorithm[29] and obtain the reconstructed attractor, seen in Fig. 5a. The scatter plot in Fig. 5a is just like the attractor of random series. With an infinitely long time series, the scatter plot

will fill the phase space like time series of white noise do. Along the orbits of the reconstructed attractor, any two points that are close to each other are likely to be far apart at the next reconstruct time step. The orbital divergence shows explosive trajectories change rather than gradual growth. We use the Grassberger–Procaccia algorithm[30,31] to calculate the correlation dimensions of the attractors of the time series. The chaotic time series was obtained by sampling a waveform from a chaotic OEO, and the random time series was obtained by sampling a waveform from the broadband random OEO. The correlation dimension results can be seen in Fig. 5b, showing good agreement with the theory.

We compare and summarize the differences between conventional single-frequency OEO, chaotic OEO and the broadband random OEO, as shown in Table 1. Traditional single-frequency OEO produces single-frequency low-phase noise signals. The advantage of traditional OEO lies in the use of low-loss characteristics of optical fiber and photoelectric feedback loop to generate ultra-low-phase noise single-frequency signals. Thus, the single-frequency OEO produces a deterministic, periodic signal. Many attempts to implement tunable OEO devices have also been reported. The major feature of the proposed broadband random OEO is that it breaks through the limitation of the cavity length of the traditional single-frequency OEO on the oscillation frequency. The proposed broadband random OEO is based on the multi-cavity length and phase matching to make the signal obtain enough energy to oscillate, instead of generating a broadband signal through nonlinear dynamics in the chaotic OEO. It leverages on the random distributed feedback in the fiber to realize the oscillation of the broadband frequencies, and the signals inherit randomness characteristics from Rayleigh scattering. The broadband random signals are more like broadband random noise than a single-frequency stable signal. On the other hand, the signal generated in the chaotic OEO is deterministic and aperiodic, and broadband. Although a chaotic signal seems to have non-period, irregularity, and noise-like characteristics, it is a deterministic signal generated by a deterministic system. We can set the same system parameters to generate exactly the same chaotic signal. This is the essential difference between the chaotic signal and the random signal generated by the broadband random OEO. The broadband random OEO generates completely different random signals every time, and it is impossible to generate the same signal by setting the same system parameters.

This proposed OEO is completely different from single-frequency OEO and chaotic OEO, and the generated signals are more complex. It is inappropriate to measure the phase noise for the complex signals it generates. Therefore, we consider evaluating signal quality from a different perspective, rather than measuring the phase noise of the signals. First, it is necessary to measure the randomness of the signals by calculating the phase space, correlation dimension, or probability density function of the signal waveform. Because the randomness is the biggest feature of this broadband random OEO scheme. Secondly, we can also measure its ability to generate large bandwidth signals, whether the power spectral density of the generated signal is flat, and whether its autocorrelation characteristics are close to white noise. The random signals should have a flat power spectrum and autocorrelation characteristics close to the delta function.

The distributed feedback in the fiber utilized by this OEO is similar in principle to the structure proposed in ref. [32], but it is also quite different. The principle of distributed feedback in the optical path has been applied in both experimental structures. However, the latter uses linear chirped fiber Bragg grating (LCFBG) as the time-delayed feedback structure of the OEO, providing a distributed feedback function. In the LCFBG, the Bragg wavelength varies along the grating, and the wavelength of

light reflected by the LCFBG is different at each location. This means that different wavelengths of light experience different delays, thus forming an equivalent broadband microwave photonics filter, which helps to generate broadband chaotic signals in the loop. A phase modulator is used in the OEO to realize electro-optic modulation, and the phase modulation–intensity modulation transition is realized due to the different delays of the different wavelengths of light. While in our experiment, the Rayleigh backscattering at any position in the optical fiber will reflect all light waves (including the light carrier and the modulated light), without selecting the reflected light wavelength. The broadband signal generated in this experiment inherits the random characteristics derived from Rayleigh scattering. Besides, the modulation method is also different from the OEO in ref. [32], intensity modulation is used in our experiment. The OEO of the two structures produced broadband chaotic signals and broadband random signals, respectively. The ability to generate broadband random signals in OEO has application potential in many scenarios. For example, it can be used as a noise source in a noise radar system. By transmitting this broadband random signal and autocorrelating it with the received echo signal, the distance to the target can be detected. Noise radar will have higher range resolution due to the larger bandwidth of the signal generated in the broadband random OEO. This random signal has a low probability of interception and is difficult to be detected and interfered.

## Discussion
The random distributed feedback based on Rayleigh backscattering combined with an OEO structure gives the birth to the device with unique characteristics. To the authors' knowledge, this is the first demonstration of an OEO capable of oscillating with an open cavity configuration and producing broadband random microwave signals. The traditional OEO only generates a single-frequency microwave signal limited by the closed cavity length. The broadband random OEO has an open cavity in the longitudinal direction, which allows the OEO to generate broadband random microwave signals by utilizing the random Rayleigh scattering in the fiber as feedback. The proposed scheme expands the capabilities of the OEO and provides a solution for OEO to generate random noise signals. The proposed OEO scheme has great potential applications in many fields. Photonic ultra-wideband signal generation, noise radar systems, random bit generation, target detection and imaging, electromagnetic interference, and secure communication are direct applications that could benefit from the broadband random OEO scheme demonstrated in this work. In particular, ultra-wideband random signals will enhance the detection of targets to higher resolution in noise radar systems, while also having the advantage of concealment. Narrow-band random signals can also be used for encryption of different channels.

## Methods
**Configuration**. The broadband random OEO was implemented using commercially available optoelectronic components. The LD is a Yenista TUNICS T100S-HP laser source, providing a signal lightwave with a wavelength of 1550 nm. The output power of the signal light source is as low as −10 dBm to suppress the Stokes light generated through SBS in the DCF (Actually, phase modulation is a commonly used effective method to suppress SBS. More details can be seen in the supplementary material Note 2.), since if the power of the Stokes light is relatively large, then it will compete with the signal light, extracting a large amount of Raman gain, which would suppress the energy of the Rayleigh backscattered signal light. The MZM is an EOspace intensity modulator with a bandwidth of 40 GHz. The Raman pump laser is a KEOPSYS S-band Raman fiber laser, providing a pump lightwave with a wavelength of 1455 nm and output power of 27 dBm. The signal light and pump light are coupled into the DCF through a 1455/1550 nm WDM, which is connected between the DCF and the light source. The DCF is 10-km long with an attenuation coefficient of 0.51 dB/km at 1550 nm. It is angled cleaved at the

open-end facet to eliminate Fresnel reflection so that all the feedback comes from the Rayleigh scattering. The Rayleigh backscattered light are filtered by a narrow-band optical filter at port 3 of the optical circulator, in order to select the signal light to get further amplification in EDFA and suppress the energy of other undesired wavelengths. A broadband PD is used to convert optical signals to radiofrequency signals. An EA is used to provide sufficient gain for the signals after the power splitter in the loop. The temporal waveforms are sampled with a real-time oscilloscope (Tektronix DPO 73304D). The operation of the broadband random OEO was also evaluated using a spectrum analyzer (Advantest R3182) to measure the spectra of the generated microwave signals, and an optical spectrum analyzer (Yokogawa AQ6370D) to measure the optical spectra.

## Data availability

The data that support the findings of this study are available from the corresponding author upon reasonable request.

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

## Acknowledgements

Thanks N. Shi and Y. Yang for comments and discussion. This work was supported by the National Key Research and Development Program of China under 2018YFB2201902 and the National Natural Science Foundation of China under 61925505. This work was also partly supported by the National Key Research and Development Program of China under 2018YFB2201901, 2018YFB2201903, and the National Natural Science Foundation of China under 61535012 and 61705217.

## Author contributions

M.L. conceived and designed the experiments, and Z.G. performed the experiments. Z.G. and T.H. conducted analytical calculations and carried out numerical simulations. Z.G., T.H., M.L., W.L., and N.Z. analyzed the data. Z.G., T.H., and J.C. wrote the paper. M.L. and N.Z. supervised the project.

## Competing interests

The authors declare no competing interests.
