## [Peer Review File · Nature Communications]

REVIEWER COMMENTS

Reviewer #1 (Remarks to the Author):

This paper describes a research study based on the realization and characterization of a new type of Opto-electronic Oscillator (OEO). The authors have assembled a new configuration of OEO whereby the Rayleigh-scattered portion of light in a long fiber is used to drive the modulator that closes the OEO loop. They refer to this configuration as "open cavity OEO" in the sense that the output of the long fiber is not fed back to the modulator, and thus does not constitute a fiber loop delay of cavity. This "open cavity" architecture is used by the authors to demonstrate generation of ultra-wideband (up to 40 GHz from DC) random microwave signals.

This paper is well-written and the investigation results are technically sound. The OEO configuration implemented in the investigation is indeed novel and has not been previously considered. There are a number of concerns that the authors might wish to address before this paper is published in Nature Communications.

1- To begin with, the notion of "oscillator" is usually associated with a device that has an output at a single frequency (could be tunable) with the major feature of low phase and amplitude noise and frequency stability (at short time intervals). Here, the major feature of the architecture described is that it generates signals over a wide bandwidth (presumably limited by a filter, or bandwidth of the modulator and any amplifiers). In fact, the chaotic nature of this oscillator is deemed useful in some of the applications suggested by the authors. So, at best, this is a chaotic oscillator; but in fact, the OEO loop, which typically amplifies filtered noise at the center frequency of the filter, is used to amplify wideband noise due to Rayleigh scattering. Indeed, a wideband "closed cavity" OEO has been previously implemented in a chaotic radar by simply removing the microwave filter and allowing the noise "oscillate in every mode of the fiber loop. The point is that a comparison of the features of the "OEO" with the conventional OEO is not strictly speaking proper. The conventional OEO is characterized by its very low phase noise. Here, the phase noise is not easy to consider (or measure). In fact, in conventional oscillators tunability in some is the only way to change extend the bandwidth of operation; the more wideband the tunability, the higher is the phase noise. Here, a continuous signal is generated across as much as 40 GHz, likely limited by the bandwidth of the modulator. Indeed, the device is more of a wideband microwave noise generator (as noted by the authors) than a wideband oscillator, a term usually referred to a device with a "single" frequency output, tunable over a wide bandwidth. This might be considered a matter of taste or semantics, but it is important enough and should be clearly mentioned in the manuscript, with some comments about the absolute noise level of the oscillator.

2- In figure 4-a, there is a clear modulation on the signal. I am not sure what is the source of it, but it looks like it could be due to the presence of the SBS peak at 9.8 GHz and its harmonics.

3- Given that the SBS is present, and despite the measures taken by the authors to reduce the input power to minimize its effect, how would the presence of this peak cause a deviation from a "perfect" random oscillation. Would the random numbers generated by such a device be affected by the presence of this non-random noise, even at low values of laser power.

4- It would be interesting if the authors could show the performance of their oscillator when a narrow-band filter is used. Since the characteristics of Rayleigh scattering noise and noise that is the source of oscillation in the OEO are the same, one wonders if there could be a "single frequency" oscillator powered by Rayleigh noise with a narrow filter.

I recommend the authors consider the issues above and make modifications to the manuscript.

Reviewer #3 (Remarks to the Author):

In this paper, the authors proposed and demonstrated a novel optoelectronic oscillator (OEO) for generating random microwave signals by utilizing Rayleigh scattering of light occurred inside a DCF fiber. Unlike in a conventional OEO in which there are distinctive modes having specific frequencies associated with the length of OEO's optoelectronic loop, in the random OEO there are no such distinctive modes because the signals feedback into the OEO loop are from distributed Rayleigh scatterings continuously along the fiber. This random OEO is analogues to the random laser first reported a few years ago, except here the generated random signals are in the microwave range. It is worth to point out that the same group of authors have recently reported several novel OEO configurations mimicking some well-known laser configurations, such as the Fourier domain mode-locked (FDML) OEO (analogues to FDML laser), optoelectronic parametric oscillator (analogues to laser parametric oscillator), and this random OEO. Undoubtedly, such efforts are fruitful, effectively shifting related physical processes from the optical domain to the microwave domain, expecting to open new doors not only in practical applications but also in understanding the physics involved. This manuscript is clearly written, and the work is original and represents a significant advancement of OEO. I recommend the publication of this paper in Nature Communications, provided that the following questions or concerns can be satisfactorily addressed.

- 1) The authors mentioned several times that the random OEO has many potential applications. Please describe a concrete example of such an application and explain why the random OEO is advantageous for the application.
- 2) The authors mentioned that the random OEO can be used in secure communication systems for encryption of different channels using the random nature of the random OEO (line 285). However, for the encryption to be meaningful, can the authors describe how to decrypt the encrypted messages? How can the intended receiver get the "key" to decrypt the random signal and how does the "key" work? I recall that random lasers were claimed to be able to encrypt messages for secure communications, but unfortunately I still have not heard a valid way for the decryption.
- 3) In line 258, chaotic OEO was mentioned, however, without describing or referencing it. It will be beneficial to the readers if a short description and related reference can be given.
- 4) In line 217, it is unclear what the sentence "The drum-shape of the noise floor..." was referred to. Which figure is to look at?
- 5) In Fig. 4b, what is the reason for the center peak and the two small pedestals symmetrically located at the two sides of the center peak?
- 6) In Fig. 4d, why did the autocorrelation show negative value around zero? The authors are also suggested to include an inset to show the expanded view of the center peak.
- 7) In line 148 and Fig. 2a, it is suggested to include the RF filter frequency response curve in Fig. 2a to convincingly explain the cause of the ramp. In general, the frequency response of a RF filter is quite flat in the pass-band.
- 8) Is there an optimized length for the DCF fiber? If yes, what determines the optimized length?
- 9) In line 171, please explain the reasons why the total cavity delay was assumed to be between 6×10^{-6} s and 6×10^{-8} s in the simulation? The DCF used has a length of 10 km, corresponding to a time delay of 50×10^{-6} s. Why was the maximum cavity delay assumed to be only 6×10^{-6} s in the simulation?
- 10) To reduce SBS threshold, it is a common practice to phase modulate the input laser to allow more laser power be injected into the fiber without SBS. In line 358 the authors mentioned that the input laser power was kept low to avoid SBS. What's the reason not to phase modulate the input laser in order to increase the input power?
- 11) Please comment on how to characterize the performance of a random OEO? In other words, what parameters or specifications are most important for the characterization of a random OEO?
- 12) Line 27, the subjective word "immense" should be removed or substituted because it is questionable whether the applications can ever be able to reach the level of "immense".

X. Steve Yao

Reviewer #4 (Remarks to the Author):

Brief Summary: The authors demonstrate the use of an open optical loop oscillator structure for generation

of wideband random microwave signals. By using distributed Rayleigh scattering in a length of optical fiber, the cavity length which is typically fixed in a conventional optoelectronic oscillator (OEO) takes a continuum of values. This allows the oscillator to produce wideband random waveforms that do not show the periodic spectral structure present in the output of typical OEOs. The work demonstrates a new and interesting method for generation of broadband signals and the data are convincing. I believe the results would be of interest to other researchers, particularly those interested in random signals for sensor or communications applications.

I believe the paper is well written and understandable. I do, however, have several comments I believe should be addressed in a revision. These are given below :

(1) While the architecture does oscillate, I would de-emphasize discussion of phase noise in the manuscript. While conventional OEOs can exhibit excellent phase noise performance, I don't believe the concept of phase noise is relevant here given the wideband random nature of the oscillations (I believe it would actually be quite poor at any given frequency).

(2) The work covered here is similar to that of Romeira et al [J. Lightwave Technol., 32 (20) 3933-3942 (2014)] that used a distributed fiber Bragg grating as the reflector. While the concept covered in the current work is new, I believe some discussion would help place the work in context.

(3) On page 5, lines 150 - 152 it is stated that the time-domain amplitude distribution agrees well with the theoretically-predicted Gaussian distribution. In my opinion, it would be useful to show the measured amplitude distribution, perhaps more so than the inset sinusoidal waveform, as this speaks to the random nature of the signal.

(4) Page 5, lines 154-155: What is the "appropriate time span" and is there any weighting applied / what is the window function used in computing the spectrogram?

(5) Throughout there is mention made to applications in communications, radar, etc. A more detailed discussion of how such waveforms might be used or how they would enable new capabilities would be useful to the reader.

Reviewer #1 (Remarks to the Author):

This paper describes a research study based on the realization and characterization of a new type of Opto-electronic Oscillator (OEO). The authors have assembled a new configuration of OEO whereby the Rayleigh-scattered portion of light in a long fiber is used to drive the modulator that closes the OEO loop. They refer to this configuration as “open cavity OEO” in the sense that the output of the long fiber is not fed back to the modulator, and thus does not constitute a fiber loop delay of cavity. This “open cavity” architecture is used by the authors to demonstrate generation of ultra-wideband (up to 40 GHz from DC) random microwave signals.

This paper is well-written and the investigation results are technically sound. The OEO configuration implemented in the investigation is indeed novel and has not been previously considered.

There are a number of concerns that the authors might wish to address before this paper is published in Nature Communications.

1- To begin with, the notion of “oscillator” is usually associated with a device that has an output at a single-frequency (could be tunable) with the major feature of low phase and amplitude noise and frequency stability (at short time intervals). Here, the major feature of the architecture described that it generates signals over a wide bandwidth (presumably limited by a filter, or bandwidth of the modulator and any amplifiers). In fact, the chaotic nature of this oscillator is deemed useful in some of the applications suggested by the authors. So, at best, this is a chaotic oscillator; but in fact, the OEO loop, which typically amplifies filtered noise at the center frequency of the filter, is used to amplify wideband noise due to Rayleigh scattering. Indeed, a wideband “closed cavity” OEO has been previously implemented in a chaotic radar by simply removing the microwave filter and allowing the noise “oscillate in every mode of the fiber loop. The point is that a comparison of the features of the “OEO” with the conventional OEO is not strictly speaking proper. The conventional OEO is characterized by its very low phase noise. Here, the phase noise is not easy to consider (or measure). In fact, in conventional oscillators tunability in some is the only way to change extend the bandwidth of operation; the more wideband the tunability, the higher is the phase noise. Here, a continuous signal is generated across as much as 40 GHz, likely limited by the bandwidth of the modulator. Indeed, the device is more of a wideband microwave noise generator (as noted by the authors) than a wideband oscillator, a term usually referred to a device with a “single” frequency output, tunable over a wide bandwidth. This might be considered a matter of taste or semantics, but it is important enough and should be clearly mentioned in the manuscript, with some comments about the absolute noise level of the oscillator.

Response: Thank you for the helpful comments and suggestions. In this paper, we proposed a novel OEO structure, which generates a complex broadband random signal. It is indeed inappropriate to compare it only with traditional single-frequency OEO. Traditional single-frequency OEO produces single-frequency low-phase noise signals. The advantage of traditional OEO lies in the use of low-loss characteristics of optical fiber and photoelectric feedback loop to generate ultra-low phase noise single-frequency signals. We can use different experimental structures to make the frequency range of single-frequency OEO tunable to obtain stable signals of different frequencies. While the major feature of the proposed broadband random OEO is that it breaks through the limitation of the cavity length of the traditional single-frequency

OEO on the oscillation frequency. The proposed broadband random OEO is based on the multi-cavity length and phase matching to make the signal obtain enough energy to oscillate, instead of generating a broadband signal through nonlinear dynamics in a chaotic OEO. It leverages on the random distributed feedback in the fiber to realize the oscillation of the broadband frequencies, and the signals inherit randomness characteristics from Rayleigh scattering. The broadband random signals are more like broadband random noise than a single-frequency stable signal. Therefore, we consider evaluating signal quality from a new perspective, rather than measuring the phase noise of the signals. First, it is necessary to measure the randomness of the signals by calculating the phase space, correlation dimension, or probability density function of the signal waveform. Because the randomness is the biggest feature of this broadband random OEO scheme. Secondly, we can also measure its ability to generate large bandwidth signals, whether the power spectral density of the generated signal is flat, and whether its autocorrelation characteristics are close to white noise. The random signals should have a flat power spectrum and autocorrelation characteristics close to the delta function.

Different from the traditional single-frequency OEO, The structure used in the chaotic OEO is a delayed feedback system to generate nonlinear dynamic behaviors such as chaos through the nonlinear effect in the cavity. In chaotic OEO, an electro-optic modulator (EOM) is used as the core non-linear device. By changing the feedback strength, the system enters an unsteady state and generates a complex chaotic signal. Although a chaotic signal seems to have non-period, irregularity, and noise-like characteristics, it is a deterministic signal generated by a deterministic system. We can set the same system parameters to generate exactly the same chaotic signal. This is the essential difference between the chaotic signal generated by the chaotic OEO and the random signal generated by the broadband random OEO. The broadband random OEO generates completely different random signals every time, and it is impossible to generate the same signal by setting the same system parameters.

The noise of traditional single-frequency OEO mainly comes from the active devices that make up the loop, such as lasers, photodetectors and amplifiers. In experiments, lasers with low relative intensity noise (RIN), photodetectors with low shot noise and low noise amplifiers are generally used to reduce the noise floor in the link. In the broadband random OEO, there are not only the aforementioned noise sources, but also the noise caused by Rayleigh scattering. The high noise floor of the link is mainly due to the use of many active devices, such as amplifiers, which introduce additional noise. We need to use an optical amplifier to amplify the weaker Rayleigh scattered light, which raises the link noise floor. A low-noise amplifier can be used to reduce the noise floor of the link.

We have added a brief introduction to chaotic OEO and the above summary on page 7-9 of the revised manuscript, and revised the content of Table 1 in the manuscript according to your suggestions. In the table we compare the traditional single-frequency OEO, chaotic OEO and the broadband random OEO to illustrate the differences between the three types OEO. We have copied the contents of this paragraph and Table 1 below for your convenience.

“However, the non-periodic and irregular characteristics of a signal generated in the broadband random OEO is similar to chaos. Different from the traditional single-frequency OEO, another type of OEO, i. e., chaotic OEO has been used as a delayed feedback system to generate nonlinear dynamic behaviors such as chaos through the nonlinear effect in the cavity. In chaotic OEO, an

electro-optic modulator (EOM) is used as the core non-linear device. By changing the feedback strength, the system enters an unsteady state and generates a complex chaotic signal. As a deterministic signal, a chaotic signal has the characteristics of non-periodic, irregular and similar to white noise.

“We compare and summarize the differences between conventional single-frequency OEO, chaotic OEO and the broadband random OEO, as shown in Table 1. Traditional single-frequency OEO produces single-frequency low-phase noise signals. The advantage of traditional OEO lies in the use of low-loss characteristics of optical fiber and photoelectric feedback loop to generate ultra-low phase noise single-frequency signals. Thus, the single-frequency OEO produces a deterministic, periodic signal. Many attempts to implement tunable OEO devices have also been reported. The major feature of the proposed broadband random OEO is that it breaks through the limitation of the cavity length of the traditional single-frequency OEO on the oscillation frequency. The proposed broadband random OEO is based on the multi-cavity length and phase matching to make the signal obtain enough energy to oscillate, instead of generating a broadband signal through nonlinear dynamics in the chaotic OEO. It leverages on the random distributed feedback in the fiber to realize the oscillation of the broadband frequencies, and the signals inherit randomness characteristics from Rayleigh scattering. The broadband random signals are more like broadband random noise than a single-frequency stable signal. On the other hand, the signal generated in the chaotic OEO is deterministic and aperiodic, and broadband. Although a chaotic signal seems to have non-period, irregularity, and noise-like characteristics, it is a deterministic signal generated by a deterministic system. We can set the same system parameters to generate exactly the same chaotic signal. This is the essential difference between the chaotic signal and the random signal generated by the broadband random OEO. The broadband random OEO generates completely different random signals every time, and it is impossible to generate the same signal by setting the same system parameters.

This novel OEO is completely different from single-frequency OEO and chaotic OEO, and the generated signals are more complex. It is inappropriate to measure the phase noise for the complex signals it generates. Therefore, we consider evaluating signal quality from a new perspective, rather than measuring the phase noise of the signals. First, it is necessary to measure the randomness of the signals by calculating the phase space, correlation dimension, or probability density function of the signal waveform. Because the randomness is the biggest feature of this broadband random OEO scheme. Secondly, we can also measure its ability to generate large bandwidth signals, whether the power spectral density of the generated signal is flat, and whether its autocorrelation characteristics are close to white noise. The random signals should have a flat power spectrum and autocorrelation characteristics close to the delta function.”

“The high noise floor of the link is mainly due to the use of many active devices, such as amplifiers, which introduce additional noise. We need to use an optical amplifier to amplify the weaker Rayleigh scattered light, which raises the link noise floor. A low-noise amplifier can be used to reduce the noise floor of the link.”

Classification	Conventional	Chaotic OEO	Broadband Random OEO
----------------	--------------	--------------------	----------------------

	single-frequency OEO		
Feedback mechanism	Closed loop	Closed loop	Open loop and Rayleigh scattering
Signal Classification	Deterministic and periodic signal	Deterministic and aperiodic signal	Stochastic and aperiodic signal
Oscillation frequency range	Single frequency (in the tunable range)	Broadband	Broadband (All frequencies within the passband)
Signal complexity	Simple	Complex	Complex

Table 1. The difference between the conventional single-frequency OEO, the chaotic OEO and the novel broadband random OEO.

2- In figure 4-a, there is a clear modulation on the signal. I am not sure what is the source of it, but it looks like it could be due to the presence of the SBS peak at 9.8 GHz and its harmonics.

Response: Thank you for the helpful comment. The peak in the noise floor in Fig. 4a is indeed due to the SBS frequency shift. From the spectrum in Fig. 4b, we can see that there is a small peak on the left and right sides of the carrier, which is the Stokes light and anti-Stokes light generated by the SBS process. We revised the description in the manuscript and added a note in Fig. 4b for clearer explanation.

The revised manuscript on page 7 and Fig. 4b are copied as follows for your convenience:

“A narrow peak at ~ 9.8 GHz is shown in the noise floor in Fig. 4a due to the Stokes shift of the stimulated Brillouin scattering (SBS). This peak disappears in the microwave signal spectrum, because spectral broadening increases the threshold of SBS²⁴. The spectral broadening is displayed in Fig. 4b. The pedestal in the lower part of the spectrum in Fig.4b is caused by the narrowband optical filter used in the experiment to filter out other wavebands. The two small peaks on the left and right sides of the optical carrier in the red line are the Stokes light and the anti-Stokes light generated by the SBS process. It can be seen from Fig. 4b that the optical spectrum is significantly broadened when the loop is closed, and all frequencies of the broad spectrum are modulated onto the signal light.”

Fig. 4(b) Optical spectrum measured after the EDFA. Blue: optical spectrum in a closed loop. Red: optical spectrum in open loop (without feedback to the MZM). The optical spectrum is significantly broadened when the loop is closed, and all frequencies of the broad spectrum are

modulated onto the signal light.

3- Given that the SBS is present, and despite the measures taken by the authors to reduce the input power to minimize its effect, how would the presence of this peak cause a deviation from a “perfect” random oscillation. Would the random numbers generated by such a device be affected by the presence of this non-random noise, even at low values of laser power.

Response: Thank you for the helpful comments. The state of the system can be observed from the phase space. In mathematics and physics, phase space is a space used to represent all possible states of a system; each possible state of the system has a corresponding point in the phase space. For random signals, the phase space shows evenly distribution of scattered points; when the signal deviates from a perfect random oscillation, the system state in the phase space will show a certain trend distribution. For example, the system state distribution in the phase space of a single-frequency signal shows a closed loop trajectory. The phase space of the chaotic signal presents a fractal structure.

As shown in Fig. S1, we use an electrical filter with a bandwidth of 8-12 GHz in the loop, and measure the waveform and spectrum in two cases. Case I: No SBS is observed in the spectrum. The signal spectrum is shown in Fig. S1a. And the reconstructed phase space displays an evenly scatter plot in Fig. S1b. Case II: There is a significant SBS frequency shift at 9.8GHz in the spectrum shown in Fig. S1c. It can be seen from the reconstructed phase space in Fig. S1d that the system state deviates from the evenly distributed scatter pattern, showing a certain trend. When the power of the SBS frequency shift component is higher, the system state deviation is more obvious. Consider an extreme situation, that is, when the SBS frequency shift power is dominant, the phase space will transition from the scattered point distribution to the closed circle distribution. When the noise power is lower, the closed circle is closer to a perfect circle. When the system deviates from a perfect random oscillation, this trend of the phase space will be reflected to the waveform. Even small deviations can be seen from the phase space. The random numbers generated by such a device will also be affected by non-random factors and may not be regarded as a perfect random number.

We have added the above discussions to part 2 of the supplement information. The revisions in the supplement information are copied as follows for your convenience.

Fig. S1 Spectrum (a) and phase space (b) results when SBS is not present. Spectrum (c) and phase space (d) results in the presence of SBS.

“2. The influence of SBS on signal randomness and the suppression method

If SBS is not completely suppressed, the randomness of the generated signal will be affected. An experiment is performed to compare the system state when there is a strong SBS and when there is no SBS.

The state of the system can be observed from the phase space. In mathematics and physics, phase space is a space used to represent all possible states of a system; each possible state of the system has a corresponding point in the phase space. For random signals, the phase space shows evenly distribution of scattered points; when the signal deviates from a perfect random oscillation, the system state in the phase space will show a certain trend distribution. For example, the system state distribution in the phase space of a single-frequency signal shows a closed loop trajectory. The phase space of the chaotic signal presents a fractal structure.

As shown in Fig. S1, we use an electrical filter with a bandwidth of 8-12 GHz in the loop, and measure the waveform and spectrum in two cases. Case I: No SBS is observed in the spectrum.

The signal spectrum is shown in Fig. S1a. And the reconstructed phase space displays an evenly scatter plot in Fig. S1b. Case II: There is a significant SBS frequency shift at 9.8GHz in the spectrum shown in Fig. S1c. It can be seen from the reconstructed phase space in Fig. S1d that the system state deviates from the evenly distributed scatter pattern, showing a certain trend. When the power of the SBS frequency shift component is higher, the system state deviation is more obvious. Consider an extreme situation, that is, when the SBS frequency shift power is dominant, the phase space will transition from the scattered point distribution to the closed circle distribution. When the noise power is lower, the closed circle is closer to a perfect circle. When the system deviates from a perfect random oscillation, this trend of the phase space will be reflected to the waveform. Even small deviations can be seen from the phase space. The random numbers generated by such a device will also be affected by non-random factors and may not be regarded as a perfect random number.”

4- It would be interesting if the authors could show the performance of their oscillator when a narrow-band filter is used. Since the characteristics of Rayleigh scattering noise and noise that is the source of oscillation in the OEO are the same, one wonders if there could be a “single-frequency” oscillator powered by Rayleigh noise with a narrow filter.

Response: Thank you for the helpful suggestion. We supplemented the experiment based on your suggestion, and used an ultra-narrow electrical filter in the loop to observe the spectrum result. The spectrum result when an ultra-narrow filter is used in the link is placed in part 4 of supplement information. For the convenience of review, we put the result in Fig. S4. We used a narrow-band electrical filter with a center frequency of 10 GHz and a 3-dB bandwidth of 29 MHz. This is the narrowest electrical filter in our laboratory. It can be seen from the spectrum that all frequencies in the passband of the ultra-narrow filter can also oscillate in the loop. Therefore, no matter how narrow the filter bandwidth is, all frequencies in its passband can get enough gain and oscillate in the loop. Since electrical filters have a certain bandwidth, it is impossible to generate single-frequency signals only by replacing a narrower filter.

We have added this content to the part 4 of the supplementary information and copied it for your convenience.

Fig. S4 Spectrum result when a narrow band filter is used in the OEO.

“4. Random signal spectrum with ultra-narrow bandwidth

By replacing the filter with this structure, all frequencies in the passband range can be vibrated, regardless of the filter bandwidth. In the case of using an ultra-narrowband filter, the single-frequency signal will not be filtered out due to the role of the filter. A narrow-band electrical filter with a center frequency of 10 GHz and a 3-dB bandwidth of 29 MHz is used in the loop. The spectrum is shown in Fig. S4. It can be seen from the spectrum that all frequencies within the passband of the filter can oscillate in the loop. Therefore, no matter how narrow the filter bandwidth is, all frequencies in its passband can get enough gain and oscillate in the loop. Single-frequency oscillation cannot be achieved simply by changing a narrower filter.”

I recommend the authors consider the issues above and make modifications to the manuscript.

Reviewer #3 (Remarks to the Author):

In this paper, the authors proposed and demonstrated a novel optoelectronic oscillator (OEO) for generating random microwave signals by utilizing Rayleigh scattering of light occurred inside a DCF fiber. Unlike in a conventional OEO in which there are distinctive modes having specific frequencies associated with the length of OEO's optoelectronic loop, in the random OEO there are no such distinctive modes because the signals feedback into the OEO loop are from distributed Rayleigh scatterings continuously along the fiber. This random OEO is analogous to the random laser first reported a few years ago, except here the generated random signals are in the microwave range. It is worth to point out that the same group of authors have recently reported several novel OEO configurations mimicking some well-known laser configurations, such as the Fourier domain mode-locked (FDML) OEO (analogous to FDML laser), optoelectronic parametric oscillator (analogous to laser parametric oscillator), and this random OEO. Undoubtedly, such efforts are fruitful, effectively shifting related physical processes from the optical domain to the microwave domain, expecting to open new doors not only in practical applications but also in understanding the physics involved. This manuscript is clearly written, and the work is original and represents a significant advancement of OEO. I recommend the publication of this paper in Nature Communications, provided that the following questions or concerns can be satisfactorily addressed.

1) The authors mentioned several times that the random OEO has many potential applications. Please describe a concrete example of such an application and explain why the random OEO is advantageous for the application.

Response: Thank you for the helpful suggestion. We have supplemented the example on noise radar on page 10 of the revised manuscript. The broadband random OEO can be used as a noise source in noise radar scheme. Noise radars detect targets by emitting random noise signals. The signal emitted by the antenna is recorded and correlated with the echo signal reflected by the target. The peak position in the autocorrelation result represents the round-trip time between the transmitted signal and the target. Since the emitted noise signal is random and has no period, it is difficult to be intercepted and interfered by the enemy, with strong concealment and anti-interference. The range detection accuracy of noise radar is related to the bandwidth of the transmitted signal. A larger bandwidth would result in a higher detection accuracy. Therefore, the broadband random OEO proposed in this paper can be used to generate ultra-wideband random signals for use in noise radars with high detection accuracy.

We have copied the revisions in the manuscript are copied here for your convenience.

"The ability to generate broadband random signals in OEO has application potential in many scenarios. For example, it can be used as a noise source in a noise radar system. By transmitting this broadband random signal and autocorrelating it with the received echo signal, the distance to the target can be detected. Noise radar will have higher range resolution due to the larger bandwidth of the signal generated in the broadband random OEO. This random signal has a low probability of interception and is difficult to be detected and interfered."

2) The authors mentioned that the random OEO can be used in secure communication systems for encryption of different channels using the random nature of the random OEO (line 285). However,

for the encryption to be meaningful, can the authors describe how to decrypt the encrypted messages? How can the intended receiver get the “key” to decrypt the random signal and how does the “key” work? I recall that random lasers were claimed to be able to encrypt messages for secure communications, but unfortunately I still have not heard a valid way for the decryption.

Response: Thank you for the helpful comment. We have a preliminary idea, that is, the device is placed at a distance between the two communicating parties, and the same random signal can be obtained by precisely controlling the delay between the two parties to the device. The sender uses the random signal to hide the information to encrypt it and deliver it to the receiver. After the receiver receives the signal, it can decrypt and obtain the information by strictly synchronizing the delay and power of the encrypted signal and the random signal, and performing differential processing on the two. It might be difficult for eavesdroppers to accurately control the time delay, so the random signal obtained cannot be differentially processed with the encrypted information to decrypt the information. This method may be used for secure communication, but it also has the risk of eavesdropping. We hope to find a safer way to prevent eavesdropping in the follow-up work.

We have added the description of the application on secure communication to part 6 of the supplement information. The content is copied here for your convenience.

*“6. The application on secure communications of the broadband random OEO
In addition to the application in the noise radar mentioned in the text, the wideband random OEO can also be used in scenarios such as chaotic communication. The random signal generated by the proposed OEO can be used as a key to encrypt information. The device can be placed at a distance between the two communicating parties, and the same random signal can be obtained by precisely controlling the delay between the two parties to the device. The sender uses the random signal to hide the information to encrypt it and deliver it to the receiver. After the receiver receives the signal, it can decrypt and obtain the information by strictly synchronizing the delay and power of the encrypted signal and the random signal, and performing differential processing on the two. It might be difficult for eavesdroppers to accurately control the time delay, so the random signal obtained cannot be differentially processed with the encrypted information to decrypt the information. This method may be used for secure communication, but it also has the risk of eavesdropping. We hope to find a safer way to prevent eavesdropping in the follow-up work.”*

3) In line 258, chaotic OEO was mentioned, however, without describing or referencing it. It will be beneficial to the readers if a short description and related reference can be given.

Response: Thank you for the helpful suggestion. We have added a brief introduction to chaotic OEO on page 8 of the revised manuscript. The OEO structure is used as a delayed feedback system in the chaotic OEO to generate nonlinear dynamic behaviors such as chaos through the nonlinear effect in the cavity. In chaotic OEO, an electro-optic modulator (EOM) is used as the core non-linear device. By changing the feedback strength, the system enters an unsteady state and generates a complex chaotic signal. The chaotic signal has the characteristics of non-periodic, irregular and similar to white noise. We have copied the brief introduction here for your convenience.

“However, the non-periodic and irregular characteristics of a signal generated in the broadband

random OEO is similar to chaos. Different from the traditional single-frequency OEO, another type of OEO is used as a delayed feedback system to generate nonlinear dynamic behaviors such as chaos through the nonlinear effect in the cavity^{26-28,32}. In chaotic OEO, an electro-optic modulator (EOM) is used as the core non-linear device. By changing the feedback strength, the system enters an unsteady state and generates a complex chaotic signal. The chaotic signal has the characteristics of non-periodic, irregular and similar to white noise.

4) In line 217, it is unclear what the sentence “The drum-shape of the noise floor...” was referred to. Which figure is to look at?

Response: Thank you for the helpful comment. “The drum-shape of the noise floor...” refers to the pedestal caused by the optical filter in Fig. 4b, and we have added a note to show that the pedestal is caused by the optical filter in Fig. 4b for a clear explanation. Since Raman amplification has an extremely large bandwidth and is capable of amplifying broadband signals, the narrow-band optical filter is used to suppress the out-of-band spectral power and only pass the light waves near the optical carrier. In order to avoid ambiguity, we modify this sentence. We have copied the content and Fig.4b below for your convenience.

“The pedestal in the lower part of the spectrum in Fig.4b is caused by the narrowband optical filter used in the experiment to filter out other wavebands.”

Fig. 4(b) Optical spectrum measured after the EDFA. Blue: optical spectrum in a closed loop. Red: optical spectrum in open loop (without feedback to the MZM). The optical spectrum is significantly broadened when the loop is closed, and all frequencies of the broad spectrum are modulated onto the signal light.

5) In Fig. 4b, what is the reason for the center peak and the two small pedestals symmetrically located at the two sides of the center peak?

Response: Thank you for the helpful comment. The two small peaks on the left and right sides of the optical carrier in Fig. 4b are the Stokes light and the anti-Stokes light generated by the SBS process. The SBS frequency shift is also shown by the peak in the noise floor spectrum in Fig. 4a. We have added an arrow pointing to the stokes light in Fig. 4b for clearer explanation. We have revised the description on page 7 of the manuscript and copied it and Fig. 4b as follows for your convenience:

“A narrow peak at ~ 9.8 GHz is shown in the noise floor in Fig. 4a due to the Stokes shift of the

stimulated Brillouin scattering (SBS). This peak disappears in the microwave signal spectrum, because spectral broadening increases the threshold of SBS²⁴. The spectral broadening is displayed in Fig. 4b. The pedestal in the lower part of the spectrum in Fig.4b is caused by the narrowband optical filter used in the experiment to filter out other wavebands. The two small peaks on the left and right sides of the optical carrier in the red line are the Stokes light and the anti-Stokes light generated by the SBS process. It can be seen from Fig. 4b that the optical spectrum is significantly broadened when the loop is closed, and all frequencies of the broad spectrum are modulated onto the signal light.”

Fig. 4(b) Optical spectrum measured after the EDFA. Blue: optical spectrum in a closed loop. Red: optical spectrum in open loop (without feedback to the MZM). The optical spectrum is significantly broadened when the loop is closed, and all frequencies of the broad spectrum are modulated onto the signal light.

6) In Fig. 4d, why did the autocorrelation show negative value around zero? The authors are also suggested to include an inset to show the expanded view of the center peak.

Response: Thank you for the helpful comment and suggestion. We have enlarged the autocorrelation map at the center peak and placed it as an inset in Fig. 4d. Judging from the autocorrelation result, there are almost no side lobes, indicating that our signal is very close to white noise. In many papers, the authors will take the absolute value of the autocorrelation function, so it seems that the autocorrelation function is all positive. After taking the absolute value of the autocorrelation result, it is convenient to compare the size of the side lobes. Whether to take the absolute value will not affect the result of the autocorrelation. Actually, it is normal for the autocorrelation result to have a negative value for broadband random signals when the time delay is close to the zero point, because the autocorrelation function of broadband random signals equals to the inverse Fourier transform of the power spectral density. For any broadband signal, the autocorrelation will have negative values near the zero point. The power spectral density of a random signal is used to describe the relationship between the energy characteristics of the signal and the frequency. It does not contain any phase information, only the power of the signal. The autocorrelation result is calculated based on the correlation degree of the waveform amplitude, so there may be an opposite trend in amplitude at some time delays. The broadband random signal generated by the OEO can be regarded as a signal with a rectangular power spectral density. Therefore, its autocorrelation function will show a shape similar to the sinc function. The result we calculated is consistent with the theory. A brief description has been added to the page 7. We have copied the content and the Fig. 4d as below for your convenience.

“The autocorrelation function has a maximum value at delay time $\tau=0$ and decays to zero quickly when delay time τ is slightly greater than zero. (The negative value is because the absolute value of the autocorrelation result is not taken, and in some papers, the absolute value of the autocorrelation function is taken to facilitate the comparison of side lobes.)”

Fig. 4(d) Autocorrelation function. The inset shows the expanded view of the center peak.

7) In line 148 and Fig. 2a, it is suggested to include the RF filter frequency response curve in Fig. 2a to convincingly explain the cause of the ramp. In general, the frequency response of a RF filter is quite flat in the pass-band.

Response: Thank you for the helpful suggestion. We did the experiment again and got a flatter spectrum result. The frequency response curve of the filter is measured and put as an inset in Fig. 2a. It can be seen from the figure that the response curve at the center of the filter is not flat enough. The unevenness of the frequency response curve will be further amplified by processes such as gain and mode competition in the optoelectronic link, and the unevenness will appear in the spectrum result. If a filter with a flatter frequency response is used, a random signal with a flat spectrum can be generated. We have added this description to page 7 of the revised manuscript, and copied the content and Fig. 2a below for your convenience.

“The ramp at the top of the spectrum is due to the non-uniformity in the magnitude response of the radiofrequency filter. The frequency response curve of the filter is measured and put as an inset in Fig.2a. It can be seen that the response curve at the center of the filter is not flat enough. The unevenness of the frequency response curve will be further amplified by processes such as gain and mode competition in the optoelectronic link, and the unevenness will appear in the spectrum result. If a filter with a flatter frequency response is used, a random signal with a flat spectrum can be generated.”

Fig. 2(a) Spectrum results of signals oscillating in the OEO. All frequencies in the passband oscillate in the OEO without discrete mode interval. The inset on the right shows a section of the spectrum with a span of 600 kHz, which clearly shows that there are no longitudinal modes under different spans. The inset on the left displays the response curve near the center frequency of the filter.

8) Is there an optimized length for the DCF fiber? If yes, what determines the optimized length?

Response: Thank you for the helpful comments. According to experimental parameters (including pump power, optical fiber parameters, etc.) and theoretical calculations, there is an optimal length for the DCF. However, it is difficult to calculate the accurate fiber length due to the complex progress in the fiber. or a certain signal, it starts to oscillate when its gain is greater than the loss. Therefore, we need to consider its power changes in the loop. In this experimental structure, we should focus on the power changes of the signal light when it enters and exits the DCF. All frequencies within the passband can oscillate, and for each frequency, there are many equally spaced scattering points in the fiber that meet the phase-matching condition. Therefore, the Rayleigh scattered light power reflected at the input port of the DCF is the sum of the reflected power of all these frequencies at all scattering points, which is a very complicated process. It is difficult to select the appropriate fiber length by calculating the optical power of a certain frequency in the passband. Therefore, we consider a rough calculation of an appropriate fiber length from the perspective of the changes in the gain and loss in the fiber.

Because the backward Rayleigh scattering is extremely weak, even if there is distributed amplification provided by stimulated Raman scattering in the fiber, its power is not large enough to make the signal oscillate in the loop. An additional optical amplifier is needed to amplify the backward Rayleigh scattered light in the loop. Therefore, we hope that as much backward Rayleigh scattered light as possible in the DCF can return to the loop to obtain higher optical power. It can be considered that the length of the optical fiber during which the pump light can amplify the signal light is the optimal length. As the signal light extracts the energy of the pump light and the fiber loss during the pumping process, the power of the pump light will gradually decrease. When the signal light power reaches half of the pump light power, it can be considered that the pump light can no longer amplify the signal light, that is, the pump exhaustion phenomenon occurs. According to the actual experimental parameters given in the article and the power balance model of the Raman amplification process, we have calculated a length of 15 km within which the pump light can continuously provide gain for the signal light. In this model, the

optimal fiber length is related to many factors, such as the effective cross-sectional area of the fiber, the loss of the pump light and signal light in the fiber, and the Raman gain coefficient. The larger the effective cross-sectional area of the optical fiber, the stronger the backward Rayleigh scattering. If the pump light loss is smaller, a longer fiber can be used to obtain greater backscatter power.

The above discussion is added in the part 1 of the supplement information. We have copied the content below for your convenience.

“According to experimental parameters (including pump power, optical fiber parameters, etc.) and theoretical calculations, there is an optimal length for the DCF. However, it is difficult to calculate the accurate fiber length due to the complex progress in the fiber: or a certain signal, it starts to oscillate when its gain is greater than the loss. Therefore, we need to consider its power changes in the loop. In this experimental structure, we should focus on the power changes of the signal light when it enters and exits the DCF. All frequencies within the passband can oscillate, and for each frequency, there are many equally spaced scattering points in the fiber that meet the phase-matching condition. Therefore, the Rayleigh scattered light power reflected at the input port of the DCF is the sum of the reflected power of all these frequencies at all scattering points, which is a very complicated process. It is difficult to select the appropriate fiber length by calculating the optical power of a certain frequency in the passband. Therefore, we consider a rough calculation of an appropriate fiber length from the perspective of the changes in the gain and loss in the fiber. Because the backward Rayleigh scattering is extremely weak, even if there is distributed amplification provided by stimulated Raman scattering in the fiber, its power is not large enough to make the signal oscillate in the loop. An additional optical amplifier is needed to amplify the backward Rayleigh scattering light in the loop. Therefore, we hope that as much backward Rayleigh scattered light as possible in the DCF can return to the loop to obtain higher optical power. It can be considered that the length of the optical fiber during which the pump light can amplify the signal light is the optimal length. As the signal light extracts the energy of the pump light and the fiber loss during the pumping process, the power of the pump light will gradually decrease. When the signal light power reaches half of the pump light power, it can be considered that the pump light can no longer amplify the signal light, that is, the pump exhaustion phenomenon occurs. According to the actual experimental parameters given in the article and the power balance model of the Raman amplification process, we have calculated a length of 15 km within which the pump light can continuously provide gain for the signal light. In this model, the optimal fiber length is related to many factors, such as the effective cross-sectional area of the fiber, the loss of the pump light and signal light in the fiber, and the Raman gain coefficient. The larger the effective cross-sectional area of the optical fiber, the stronger the backward Rayleigh scattering. If the pump light loss is smaller, a longer fiber can be used to obtain greater backscatter power.”

9) In line 171, please explain the reasons why the total cavity delay was assumed to be between 6×10^{-6} s and 6×10^{-8} s in the simulation? The DCF used has a length of 10 km, corresponding to a time delay of 50×10^{-6} s. Why was the maximum cavity delay assumed to be only 6×10^{-6} s in the simulation?

Response: Thank you for the helpful comments. Taking into account the forward and backward

propagation of light in DCF, the longest round-trip delay of 10 km optical fiber is 10×10^{-5} s. The shortest time is 6×10^{-8} s when the light is reflected at the input of the DCF, corresponding to the cavity length without the DCF part. Considering that all frequencies in the passband range are counted, and each frequency undergoes random scattering at a large number of scattering position in the fiber and the superposition processes of the total backscattered power, simulating this process of power changing in DCF requires a very huge amount of calculation. If the simulation is based entirely on real parameters and maintains high accuracy, the amount of calculation is extremely large, and our server memory is not large enough to support such a large amount of calculation. Therefore, in order to show the characteristics of broadband spectrum and randomness, we made a trade-off and chose a short period of time to reduce the amount of calculation. We assume that the gain is mainly provided by the amplifier. Moreover, taking into account the Raman amplification model, the backscattered light power is relatively high near the input end of the DCF. As the signal light and pump light propagate along the fiber, the Raman gain effect decreases, and the backscattered power decays rapidly. So in the simulation we only took part of the round-trip time of the fiber where the Raman gain is relative high. Based on the above reasons, we did not calculate according to the actual fiber length during the simulation. We also added an explanation to page 6 in the manuscript to explain why only this period of time was selected for calculation. The content is copied below for your convenience.

“Considering the limitation of calculation accuracy and calculation amount, we only take a shorter length of fiber for calculation in simulation, and the total cavity delay is assumed to change between 6×10^{-6} s and 6×10^{-8} s.”

10) To reduce SBS threshold, it is a common practice to phase modulate the input laser to allow more laser power be injected into the fiber without SBS. In line 358 the authors mentioned that the input laser power was kept low to avoid SBS. What’s the reason not to phase modulate the input laser in order to increase the input power?

Response: Thank you for the helpful comment and suggestion. As you said, phase modulation is a commonly used effective method to suppress SBS, and we have also tried this method in our experiments. We use the 8 Gs/s Pseudo-Random Binary Sequence (PRBS) signal to modulate the optical carrier through a phase modulator placed between the laser and the MZM, and the spectrum is broadened, thereby effectively suppressing the SBS process, as shown in the spectrum diagram in Fig. S2a. First, we disconnect the feedback electrical signal to the MZM. The spectrum is shown in red in Fig. S2, and a clear SBS frequency shift can be seen. Then we perform phase modulation on the optical signal, and the spectrum result is shown by the blue line. The spectrum of the blue line is obviously broadened, and SBS is suppressed. At the same time, we observed the noise floor obtained from the PD before and after modulation through a spectrum analyzer, as shown in Fig. S2b. The open-loop noise floor signal when no phase modulation is applied is shown in the red spectrum. When phase modulation is used, the noise floor is obviously raised from DC to several GHz. Although the spectrum is only slightly broadened, it generates several gigahertz additional noise signals when mapped in the microwave domain. Generally, when phase modulation is used, the modulated signal cannot be detected from the PD. However, the random distribution feedback in the fiber and the abnormal dispersion of the DCF will change the phase relationship of light, thus realizing the conversion from phase modulation to intensity

modulation. The PRBS signal through phase modulation is detected from the PD.

We think that readers may confuse this uplift of the noise floor caused by the application of external modulation with its own oscillation signal after the OEO feedback is closed. Therefore, in order to avoid this kind of misunderstanding, we did not adopt the phase modulation method in the experimental scheme of this article, and only suppress the SBS process by reducing the signal optical power. However, this method of phase modulation is still very effective. In subsequent experiments, the phase modulation method you proposed can be used to increase the power of the optical carrier input to the DCF. We have added a short description on page 13 in the revised manuscript and put the details in part 2 of the supplementary information. We have copied the revised manuscript below for your convenience.

Fig. S2 (a) the spectral broadening (blue) with phase modulation on the optical carrier to suppress the SBS. The red line shows spectrum of open loop without modulation. (b) Noise floor when phase modulation broadens the spectrum (blue) and when there is no modulation (red).

“The laser diode is a Yenista TUNICS T100S-HP laser source, providing a signal light wave with a wavelength of 1550 nm. The output power of the signal light source is as low as -10 dBm to suppress the Stokes light generated through SBS in the DCF (Actually, phase modulation is a commonly used effective method to suppress SBS. More details can be seen in the supplementary material.), since if the power of the Stokes light is relatively large, then it will compete with the signal light, extracting a large amount of Raman gain, which would suppress the energy of the Rayleigh backscattered signal light.”

“In order to allow more laser power be injected into the DCF and avoid the influence of SBS, we should take reasonable measures to suppress SBS. Phase modulation on the input laser is a commonly used effective method to suppress SBS. An experiment is performed with phase modulation. First, we disconnect the feedback electrical signal to the MZM (open loop). The optical spectrum is shown in red in Fig. S2a, and a clear SBS frequency shift can be seen. Then, an 8 Gs/s Pseudo-Random Binary Sequence is used to modulate the optical carrier through a phase modulator placed between the laser and the MZM, and the spectrum result is as shown by the blue line. The spectrum of the blue line is obviously broadened, and SBS is suppressed. At the same time, we observed the noise floor obtained from the PD before and after modulation through a spectrum analyzer, as shown in Fig. S2b. The open-loop noise floor signal when no phase modulation is applied is shown in the red spectrum. When phase modulation is used, the noise floor is obviously raised from DC to several GHz. Although the spectrum is only slightly broadened, it generates several GHz additional noise signals when mapped in the microwave

domain. Generally, when phase modulation is used, the modulated signal cannot be detected from the PD. However, the random distribution feedback in the fiber and the abnormal dispersion of the DCF will change the phase relationship of light, thus realizing the transition from phase modulation to intensity modulation. The PRBS signal through phase modulation is detected from the PD. In order not to confuse this uplift of the noise floor caused by the application of external modulation with its own oscillation signal after the OEO feedback, phase modulation is not adopted in the experiment. We only suppress the SBS process by reducing the signal optical power. However, this method of phase modulation is still very effective. In subsequent experiments, the phase modulation method can be used to increase the power of the optical carrier input to the DCF.”

11) Please comment on how to characterize the performance of a random OEO? In other words, what parameters or specifications are most important for the characterization of a random OEO?

Response: Thank you for the helpful comments. Since this is a novel broadband OEO, it is not suitable for measuring the phase noise of the complex signals it generates. We believe that the signal quality should be measured from a new perspective. First, it is necessary to measure the randomness of the signals by calculating the phase space, correlation dimension, or probability density function of the signal waveform. Because the randomness is the biggest feature of this broadband random OEO scheme. Secondly, we can also measure its ability to generate large bandwidth signals, whether the power spectral density of the generated signal is flat, and whether its autocorrelation characteristics are close to white noise. We have added the comment in the revised manuscript on page 9, and copied the content below for your convenience.

“This novel OEO is completely different from single-frequency OEO and chaotic OEO, and the generated signals are more complex. It is inappropriate to measure the phase noise for the complex signals it generates. Therefore, we consider evaluating signal quality from a new perspective, rather than measuring the phase noise of the signals. First, it is necessary to measure the randomness of the signals by calculating the phase space, correlation dimension, or probability density function of the signal waveform. Because the randomness is the biggest feature of this broadband random OEO scheme. Secondly, we can also measure its ability to generate large bandwidth signals, whether the power spectral density of the generated signal is flat, and whether its autocorrelation characteristics are close to white noise. The random signals should have a flat power spectrum and autocorrelation characteristics close to the delta function.”

12) Line 27, the subjective word “immense” should be removed or substituted because it is questionable whether the applications can ever be able to reach the level of “immense”.

Response: Thank you for the helpful suggestion. We have removed the word “immense” from the manuscript. The revised sentence is copied below for your convenience.

“The proposed device has potential in many fields such as random bit generation, radar systems, electronic interference and countermeasures, and telecommunications.”

Reviewer #4 (Remarks to the Author):

Brief Summary: The authors demonstrate the use of an open optical loop oscillator structure for generation of wideband random microwave signals. By using distributed Rayleigh scattering in a length of optical fiber, the cavity length which is typically fixed in a conventional optoelectronic oscillator (OEO) takes a continuum of values. This allows the oscillator to produce wideband random waveforms that do not show the periodic spectral structure present in the output of typical OEOs. The work demonstrates a new and interesting method for generation of broadband signals and the data are convincing. I believe the results would be of interest to other researchers, particularly those interested in random signals for sensor or communications applications.

I believe the paper is well written and understandable. I do, however, have several comments I believe should be addressed in a revision. These are given below:

(1) While the architecture does oscillate, I would de-emphasize discussion of phase noise in the manuscript. While conventional OEOs can exhibit excellent phase noise performance, I don't believe the concept of phase noise is relevant here given the wideband random nature of the oscillations (I believe it would actually be quite poor at any given frequency).

Response: Thank you for the helpful comments. Since this is a novel broadband OEO, it is true that measuring the phase noise for the generated complex signals is inappropriate. We believe that the signal quality should be measured from a new perspective. First, it is necessary to measure the randomness of the signals by calculating the phase space, correlation dimension, or probability density function of the signal waveform. Because the randomness is the biggest feature of this broadband random OEO scheme. Secondly, we can also measure its ability to generate large bandwidth signals, whether the power spectral density of the generated signal is flat, and whether its autocorrelation characteristics are close to white noise. We have added the comment in the revised manuscript on page 9, and have copied the content below for your convenience.

"This novel OEO is completely different from single-frequency OEO and chaotic OEO, and the generated signals are more complex. It is inappropriate to measure the phase noise for the complex signals it generates. Therefore, we consider evaluating signal quality from a new perspective, rather than measuring the phase noise of the signals. First, it is necessary to measure the randomness of the signals by calculating the phase space, correlation dimension, or probability density function of the signal waveform. Because the randomness is the biggest feature of this broadband random OEO scheme. Secondly, we can also measure its ability to generate large bandwidth signals, whether the power spectral density of the generated signal is flat, and whether its autocorrelation characteristics are close to white noise. The random signals should have a flat power spectrum and autocorrelation characteristics close to the delta function."

(2) The work covered here is similar to that of Romeira et al [J. Lightwave Technol., 32 (20) 3933-3942 (2014)] that used a distributed fiber Bragg grating as the reflector. While the concept covered in the current work is new, I believe some discussion would help place the work in context.

Response: Thank you for the helpful suggestion. We have cited this paper in the manuscript and

discussed the two works. The distributed feedback in the fiber utilized by this OEO is similar in principle to the structure proposed in Refs. [32], but it is also quite different. The principle of distributed feedback in the optical path has been applied in both experimental structures. However, the latter uses linear chirped fiber Bragg grating (LCFBG) as the time-delayed feedback structure of the OEO, providing a distributed feedback function. In the LCFBG, the Bragg wavelength varies along the grating, and the wavelength of light reflected by the LCFBG is different at each location. This means that different wavelengths of light experience different delays, thus forming an equivalent broadband microwave photonics filter, which helps to generate broadband chaotic signals in the loop. A phase modulator is used in the OEO to realize electro-optic modulation, and the phase modulation - intensity modulation transition is realized due to the different delays of the different wavelengths of light. While in our experiment, the Rayleigh backscattering at any position in the optical fiber will reflect all light waves (including the light carrier and the modulated light), without selecting the reflected light wavelength. The broadband signal generated in this experiment inherits the random characteristics derived from Rayleigh scattering. Besides, the modulation method is also different from the OEO in Refs. [32], intensity modulation is used in our experiment. The OEO of the two structures produced broadband chaotic signals and broadband random signals, respectively.

The above discussion about the two works is added on page 9-10 in the revised manuscript. We have copied the revised manuscript here for your convenience.

“The distributed feedback in the fiber utilized by this OEO is similar in principle to the structure proposed in Refs. [32], but it is also quite different. The principle of distributed feedback in the optical path has been applied in both experimental structures. However, the latter uses linear chirped fiber Bragg grating (LCFBG) as the time-delayed feedback structure of the OEO, providing a distributed feedback function. In the LCFBG, the Bragg wavelength varies along the grating, and the wavelength of light reflected by the LCFBG is different at each location. This means that different wavelengths of light experience different delays, thus forming an equivalent broadband microwave photonics filter, which helps to generate broadband chaotic signals in the loop. A phase modulator is used in the OEO to realize electro-optic modulation, and the phase modulation - intensity modulation transition is realized due to the different delays of the different wavelengths of light. While in our experiment, the Rayleigh backscattering at any position in the optical fiber will reflect all light waves (including the light carrier and the modulated light), without selecting the reflected light wavelength. The broadband signal generated in this experiment inherits the random characteristics derived from Rayleigh scattering. Besides, the modulation method is also different from the OEO in Refs. [32], intensity modulation is used in our experiment. The OEO of the two structures produced broadband chaotic signals and broadband random signals, respectively.”

(3) On page 5, lines 150 - 152 it is stated that the time-domain amplitude distribution agrees well with the theoretically-predicted Gaussian distribution. In my opinion, it would be useful to show the measured amplitude distribution, perhaps more so than the inset sinusoidal waveform, as this speaks to the random nature of the signal.

Response: Thank you for the helpful suggestion. We have added the waveform amplitude distribution result to Fig. 2b. Both the waveform and the probability density function are shown in

Fig. 2b. The random waveform with Gaussian probability density function indicates that the signal is very close to Gaussian white noise. The revised manuscript and Fig. 2b are copied here for your convenience.

“The temporal waveform of the output microwave signal from the splitter is sampled with a real-time oscilloscope at a sampling rate of 100 GS/s as shown in Fig. 2b. The amplitude varies randomly, showing a probability density function (PDF) of Gaussian distribution, as shown in the right panel in Fig.2b, which is in good agreement with the theory¹⁸. The inset shows the details of the waveform, which approximates a sinusoidal signal with the period varies slightly around 0.2 ns.”

Fig. 2(b) Temporal waveform and the probability density function of the signal. The inset is a section of the waveform. The waveform approximates a sinusoidal signal with a period that varies slightly around 0.2 ns. The right panel displays the Gaussian probability density function of the random signal.

(4) Page 5, lines 154-155: What is the "appropriate time span" and is there any weighting applied / what is the window function used in computing the spectrogram?

Response: Thank you for the helpful comments. The “appropriate time span” here actually refers to the appropriate window length. Because in the short-time Fourier transform process, the length of the window determines the time resolution and frequency resolution of the spectrogram. The length of the window determines the length of the intercepted signal. If the window length is long, the frequency resolution after Fourier transform of the intercepted signal is high, but the time resolution is poor. If the window length is short, the frequency resolution after Fourier transform of the intercepted signal is worse, and the time resolution is better. Therefore, in the short-time Fourier transform, high time resolution and high frequency resolution cannot be achieved simultaneously, and the trade-off should be made according to specific needs. In this article, in order to allow readers to distinguish more clearly the difference in power changes between different frequencies and the power changes of the same frequency at different times, we tried different window lengths, and finally selected this result to present to the readers. The hamming window is used in the short-time Fourier transform process, and there is no additional weighting. In order to avoid ambiguity, we have revised the manuscript according to the above discussion and copied the sentence here for your convenience.

“A short-time Fourier transform was performed to the sampled data. In this process, a Hamming window was selected to intercept the signal, and an appropriate window length was selected. The

result is shown in Fig.2c.”

(5) Throughout there is mention made to applications in communications, radar, etc. A more detailed discussion of how such waveforms might be used or how they would enable new capabilities would be useful to the reader.

Response: Thank you for the helpful suggestion. We have supplemented the example on noise radar on page 10 of the revised manuscript. The broadband random OEO can be used as a noise source in a noise radar system. By transmitting this broadband random signal and autocorrelating it with the received echo signal, the distance to the target can be detected. Noise radar will have higher range resolution due to the larger bandwidth of the signal generated in the broadband random OEO. This random signal has a low probability of interception and is difficult to be detected and interfered.

We have copied the content of this paragraph below for your convenience.

“The ability to generate broadband random signals in OEO has application potential in many scenarios. For example, it can be used as a noise source in a noise radar system. By transmitting this broadband random signal and autocorrelating it with the received echo signal, the distance to the target can be detected. Noise radar will have higher range resolution due to the larger bandwidth of the signal generated in the broadband random OEO. This random signal has a low probability of interception and is difficult to be detected and interfered.”

REVIEWER COMMENTS

Reviewer #1 (Remarks to the Author):

I am happy to recommend this revised version of the original paper for publications. The changes made by the authors make it a stronger paper and suitable for the publication in this Journal.

Reviewer #3 (Remarks to the Author):

The authors have addressed my concerns and suggestions reasonably well and I recommend to accept the manuscript for publication pending on some minor changes suggested below.

1) For my comment #2 on the application on secure communications of the broadband random OEO, it may be better to revise the language as follows:

In addition to the application in the noise radar mentioned in the text, the wideband random OEO can also be used in secure communications for the encryption of the information, although how to decrypt the information by the intended receiver without the risk of being eavesdropped still requires further investigation."

2) For my comment #9 on the length of DCF, the following is suggested :

Considering the huge amount computation power required for an accurate simulation, we only take the first 0.6 km of the DCF fiber in the simulation where the Ramon gain is most significant, corresponding to a total cavity delay between 6×10^{-6} s and 6×10^{-8} s, although a total of 10 km DCF fiber was used in the experiment.

Reviewer #4 (Remarks to the Author):

Thank you for addressing most of my comments. I still have several outstanding ones that need to be addressed further.

(1) With Figure 2 (b) and the discussion surrounding the amplitude distribution: Thank you for adding the amplitude distribution. It seems, however, the distribution is closer to a Lorentzian (possibly a Voight) distribution than a Gaussian. I suggest adding the theoretical distribution to the plot and discussing the fit accuracy quantitatively in the text.

(2) With respect to the window function for the spectrograms: I understand how choosing the window works, what I believe is necessary is a numeric value for its duration.

Reviewer #1 (Remarks to the Author):

I am happy to recommend this revised version of the original paper for publications. The changes made by the authors make it a stronger paper and suitable for the publication in this Journal.

Response: Thanks for your review. Your suggestions are very important to the improvement of this paper. We would like to take this opportunity to thank you again for the efforts and valuable suggestions.

Reviewer #3 (Remarks to the Author):

The authors have addressed my concerns and suggestions reasonably well and I recommend to accept the manuscript for publication pending on some minor changes suggested below.

1) For my comment #2 on the application on secure communications of the broadband random OEO, it may be better to revise the language as follows:

In addition to the application in the noise radar mentioned in the text, the wideband random OEO can also be used in secure communications for the encryption of the information, although how to decrypt the information by the intended receiver without the risk of being eavesdropped still requires further investigation."

Response: Thank you for the helpful suggestion. We have revised the content in part 6 of the supplement information according to your advice.

2) For my comment #9 on the length of DCF, the following is suggested:

Considering the huge amount computation power required for an accurate simulation, we only take the first 0.6 km of the DCF fiber in the simulation where the Ramon gain is most significant, corresponding to a total cavity delay between 6×10^{-6} s and 6×10^{-8} s, although a total of 10 km DCF fiber was used in the experiment.

Response: Thank you for the helpful suggestion. We have also revised the content on page 7 of the revised manuscript according to your advice.

Reviewer #4 (Remarks to the Author):

Thank you for addressing most of my comments. I still have several outstanding ones that need to be addressed further.

(1) With Figure 2 (b) and the discussion surrounding the amplitude distribution: Thank you for adding the amplitude distribution. It seems, however, the distribution is closer to a Lorentzian (possible a Voigt) distribution than a Gaussian. I suggest adding the theoretical distribution to the plot and discussing the fit accuracy quantitatively in the text.

Response: Thank you for the helpful comments and suggestions. The calculated probability density function is not a perfect Gaussian distribution and has a certain deviation. The calculated mathematical expectation μ is -7.9×10^{-3} , and the calculated standard deviation σ is 0.1451.

We also calculated the ideal Gaussian distribution, and calculated the correlation coefficient between the probability density function and the ideal Gaussian distribution. The value of the correlation coefficient R is 0.9979. The deviation of the probability density distribution from an ideal Gaussian distribution may be caused by the noise introduced by other devices in the link and the unevenness of the oscillation signal in the passband. We put the probability density function together with an ideal Gaussian distribution and Lorentz distribution for comparison, as shown in the Fig. S1. It can be seen that the probability density function is closer to the Gaussian distribution. The reason it looks more like the Lorentz distribution is that there are visual errors since we shrank the picture of the probability density function in original Fig. 2b. We have readjusted the scale of the probability density function graph in Fig.2b to avoid this visual error, and added the ideal Gaussian distribution curve (orange dotted line) to the graph. We have added the above discussions on page 5 of the revised manuscript. The revisions in the manuscript are copied as follows for your convenience.

Fig. S1 The probability density function (blue), the ideal Gaussian distribution curve (orange) and the ideal Lorentz distribution (green).

“The amplitude varies randomly, showing a probability density function (PDF, blue solid line) of Gaussian distribution, with the mathematical expectation μ is -7.9×10^{-3} , and the standard deviation σ is 0.1451, as shown in the right panel in Fig.2b. We also calculated the ideal Gaussian distribution (orange dotted line), and calculated the correlation coefficient between the probability density function and the ideal Gaussian distribution. The value of the correlation coefficient R is 0.9979. The deviation of the probability density distribution may be caused by the noise introduced by other devices in the link and the unevenness of the oscillating signal in the passband. The experimental results are relatively consistent with the theory¹⁸.”

The revised Fig.2b Temporal waveform and the probability density function of the signal. The inset is a section of the waveform. The waveform approximates a sinusoidal signal with a period that varies slightly around 0.2 ns. The right panel displays the Gaussian probability density function of the random signal as shown by the blue solid line. The orange dotted line shows the Gaussian distribution. The mathematical expectation μ is -7.9×10^{-3} , and the standard deviation σ is 0.1451. The correlation coefficient R is 0.9979.

(2) With respect to the window function for the spectrograms: I understand how choosing the window works, what I believe is necessary is a numeric value for its duration.

Response: Thank you for the helpful suggestion. The duration of the Hamming window in the short-time Fourier transform is set to 6 μs . Two adjacent windows have an intersection, and the length of the overlapping part is set to 3 μs . We have added the duration of the window in the content on page 5 of the revised manuscript. The revisions in the manuscript are copied as follows for your convenience

“A short-time Fourier transform was performed to the sampled data. In this process, a Hamming window was selected to intercept the signal, and an appropriate window length was selected. In this paper, the window length is set to 6 μs . Two adjacent windows have an intersection, and the length of the overlapping part is set to 3 μs .”

REVIEWERS' COMMENTS

Reviewer #4 (Remarks to the Author):

Thank you for addressing my comments. I find the revised manuscript suitable for publication in Nature Communications.

Reviewer #4 (Remarks to the Author):

Thank you for addressing my comments. I find the revised manuscript suitable for publication in Nature Communications.

Response: Thank you for the positive comments and recommendation. We would like to take this opportunity to thank you again for the efforts and valuable suggestions.